# Clathrin light chain A drives selective myosin VI recruitment to clathrin-coated pits under membrane tension

Matteo Biancospino[1,6], Gwen R. Buel [2,6], Carlos A. Niño[1,6], Elena Maspero[1], Rossella Scotto di Perrotolo[1], Andrea Raimondi [3], Lisa Redlingshöfer[4], Janine Weber[1], Frances M. Brodsky[4]*, Kylie J. Walters[2]* & Simona Polo [1,5]*

Clathrin light chains (CLCa and CLCb) are major constituents of clathrin-coated vesicles. Unique functions for these evolutionary conserved paralogs remain elusive, and their role in clathrin-mediated endocytosis in mammalian cells is debated. Here, we find and structurally characterize a direct and selective interaction between CLCa and the long isoform of the actin motor protein myosin VI, which is expressed exclusively in highly polarized tissues. Using genetically-reconstituted Caco-2 cysts as proxy for polarized epithelia, we provide evidence for coordinated action of myosin VI and CLCa at the apical surface where these proteins are essential for fission of clathrin-coated pits. We further find that myosin VI and Huntingtin-interacting protein 1-related protein (Hip1R) are mutually exclusive interactors with CLCa, and suggest a model for the sequential function of myosin VI and Hip1R in actin-mediated clathrin-coated vesicle budding.

[1] IFOM, Fondazione Istituto FIRC di Oncologia Molecolare, 20139 Milan, Italy. [2] Structural Biophysics Laboratory, Center for Cancer Research, National Cancer Institute, Frederick, MD 21702, USA. [3] Experimental Imaging Center, San Raffaele Scientific Institute, Milan, Italy. [4] Division of Biosciences, University College London, London WC1E 6BT, UK. [5] Dipartimento di Oncologia ed Emato-oncologia, Universita' degli Studi di Milano, 20122 Milan, Italy. [6]These authors contributed equally: Matteo Biancospino, Gwen R. Buel, Carlos A. Niño. *email: f.brodsky@ucl.ac.uk; kylie.walters@nih.gov; simona.polo@ifom.eu

Clathrin-coated vesicles (CCVs) are well-characterized molecular machines, responsible for the internalization of nutrients, signaling receptors, and adhesion molecules from the plasma membrane (PM) as well as for cargo transport between the *trans*-Golgi network and endosomes[1,2]. Clathrin is a three-legged molecule (triskelion), composed of three clathrin heavy chain (CHC) and three clathrin light chain (CLC) subunits. The clathrin triskelia assemble around a budding vesicle, creating a characteristic polyhedral coat, forming a clathrin-coated pit (CCP) that captures protein cargo and further matures into a CCV.

While CHCs are essential components of the cages, CLCs are apparently more regulatory. Functional roles for CLCs have been linked to regulation of cage rigidity, assembly and disassembly by extrapolation from in vitro studies[1], and to clathrin-mediated endocytosis (CME) of model cargos, including G-protein-coupled receptors[3–5]. Importantly, CLCs appear essential at the apical surface of highly polarized tissues, where membrane tension is higher than normal and actin dynamics is critical for membrane invagination and fission[6]. This process involves the conserved interaction between CLCs and the actin-remodeling huntingtin-interacting proteins (Hip1 and Hip1R)[7,8], an interaction that also contributes to clathrin–actin interactions during endosomal recycling and is needed for cell migration[9]. CLCs are also required for uptake of large particles, including viruses and pathogenic bacteria, such as *Listeria*, *Yersinia*, and *Salmonella*[10–12]. In these cases, the direct association of CLC with actin remodelers distinguishes these clathrin-dependent processes from canonical CME. Finally, CLCs, together with Hip1R and the actin motor protein myosin VI[13], participate in cell–cell junction formation and remodeling through molecular mechanisms that have not been defined[14].

In vertebrates, two genes encode the CLCa and CLCb variants, which are 40% divergent in sequence. High conservation of each protein variant in all vertebrate species suggests that they have distinct functions beyond their shared interaction with the Hip proteins. However, evidence for CLCa- and CLCb-specific roles is limited. CLCa was shown to be preferentially involved in cell spreading and migration[15], whereas maturation of CCPs from flat lattices to invaginated buds requires phosphorylation of CLCb[16]. CLCb was also found to be overexpressed in non-small-cell lung cancer, causing altered CCP dynamics and consequently aberrant growth factor signaling[17].

In a previous study, we identified clathrin as a major and specific binding partner of myosin VI isoforms that contain the α2-linker (named myosin VI$_{long}$) and we found that myosin VI$_{long}$ localizes to CCPs independently of the presence of the endocytic adaptor Dab-2[18]. These findings suggested a molecular link between clathrin and myosin VI$_{long}$. In this manuscript, we now demonstrate that this interaction is due to a direct and selective interaction of myosin VI$_{long}$ with CLCa and not CLCb. We use nuclear magnetic resonance (NMR) spectroscopy to solve the structure of the myosin VI$_{long}$:CLCa complex and characterize the mechanism of interaction. The CLCa-binding region in myosin VI partially overlaps with the ubiquitin-binding domain MyUb[19] while CLCa binds myosin VI with a surface that is in close proximity to the Hip1R-binding site[7]. Guided by the structure, we identify a CLCa point mutant that loses myosin VI-binding ability, while leaving unaltered its capacity to interact with CHC or Hip1R. We further investigate the biological implications of these molecular interactions, taking advantage of genetically reconstituted Caco-2 cells growing in three dimension (3D) as cysts. Altogether, our results indicate that the interaction between myosin VI$_{long}$ and CLCa is needed to generate the force that leads to invagination and fission of CCPs at the apical surface of highly polarized cells and suggests a model for the sequential function of myosin VI and Hip1R in actin-mediated coated vesicle budding.

## Results

### CLCa is a direct and specific interactor of myosin VI$_{long}$.
In myosin VI, the presence of helix α2 structurally defines a clathrin-binding domain that is unique to myosin VI$_{long}$[18,19] (Fig. 1a). To identify the corresponding myosin VI$_{long}$ interacting region within clathrin, we performed experiments with green fluorescent protein (GFP)- and glutathione S-transferase (GST)- tagged CHC truncations. The results revealed that the C-terminal part of the CHC is required for myosin VI interaction but failed to demonstrate direct binding between the two proteins (Supplementary Fig. 1a). Given that CLCs are reported to interact with the C-terminus of CHC, we hypothesized their involvement in recruiting myosin VI to CCPs. Corroborating our hypothesis, confocal microscopic analysis of HeLa cells revealed reduced co-localization of myosin VI$_{long}$ and CHC upon CLCa transient knockdown (KD) as measured by Mander's coefficient (Supplementary Fig. 1b, c). Unexpectedly, CLCb KD had the opposite effect, increasing the co-localization levels of CHC and myosin VI$_{long}$, thus suggesting different roles for the two CLCs.

We next examined direct binding in a pull-down assay by using bacterially purified GST-CLCa and GST-CLCb full-length proteins and purified fragments of long and short myosin VI isoforms spanning amino acids 998–1131 (Fig. 1a). The results demonstrated direct and selective binding of CLCa to myosin VI$_{long}$ while both CLCs showed minimal binding to myosin VI$_{short}$ (Fig. 1b).

To formally test whether CLCa acts as a bridge between myosin VI$_{long}$ and CHC, we exploited CLCa knockout (KO) mice that were recently generated[5] and newly produced CLCb KO mice. Whole-brain lysates obtained from CLCa and CLCb KO mice and wild-type (WT) littermates were used in a pull-down assay with GST-myosin VI$^{998–1131}$ from long and short myosin VI isoforms. Using WT littermates, GST-myosin VI$_{long}$ efficiently pulled down CHC (Fig. 1c). This interaction was dependent on CLCa since clathrin was not detected using brain lysate from CLCa KO animals (Fig. 1c). Of note, brain lysate from CLCb KO mice behaved as WT littermates, similarly showing interaction between clathrin and myosin VI$_{long}$ (Fig. 1c), despite lacking CLCb (Fig. 1d).

The lysis conditions used in these experiments and in our previous study[18] favor free triskelia in solution as opposed to assembled clathrin. During clathrin cage assembly, CLCs undergo conformational changes[8], and to investigate whether the interaction of the coat protein with myosin VI can occur in the context of a formed CCV, we exploited the ability of clathrin to self-assemble in a pH-dependent manner[20,21]. We purified clathrin from pig brain CCVs and induced the formation of assembled cages from native clathrin containing an endogenous mixture of CLCa and CLCb, CHC with no CLCs, or CHC reconstituted with either CLCa or CLCb only (see Methods and Supplementary Fig. 2a–c for details). Purified myosin VI$^{998–1131}$ protein co-sedimented with native clathrin cages and cages formed from clathrin with only CLCa, whereas only weak sedimentation was observed with CLCb-only cages that was similar to CHC-only cages (Fig. 1e). In summary, these results demonstrate that CLCa is the subunit of clathrin responsible for binding to myosin VI$_{long}$, both in the context of free triskelia and in CCVs.

### CLCa and myosin VI$_{long}$ interact with sub-micromolar affinity.
To obtain biochemical and structural insights into the myosin VI$_{long}$:CLCa complex, we systematically designed CLCa deletions taking into consideration the known interaction domains of the protein[22] (Fig. 2a). Incubation of these constructs with bacterially purified myosin VI$^{998–1131}$ identified a region of interaction in CLCa that spans amino acids 47–97 (Fig. 2b). This fragment of clathrin falls in a region of limited conservation between the two

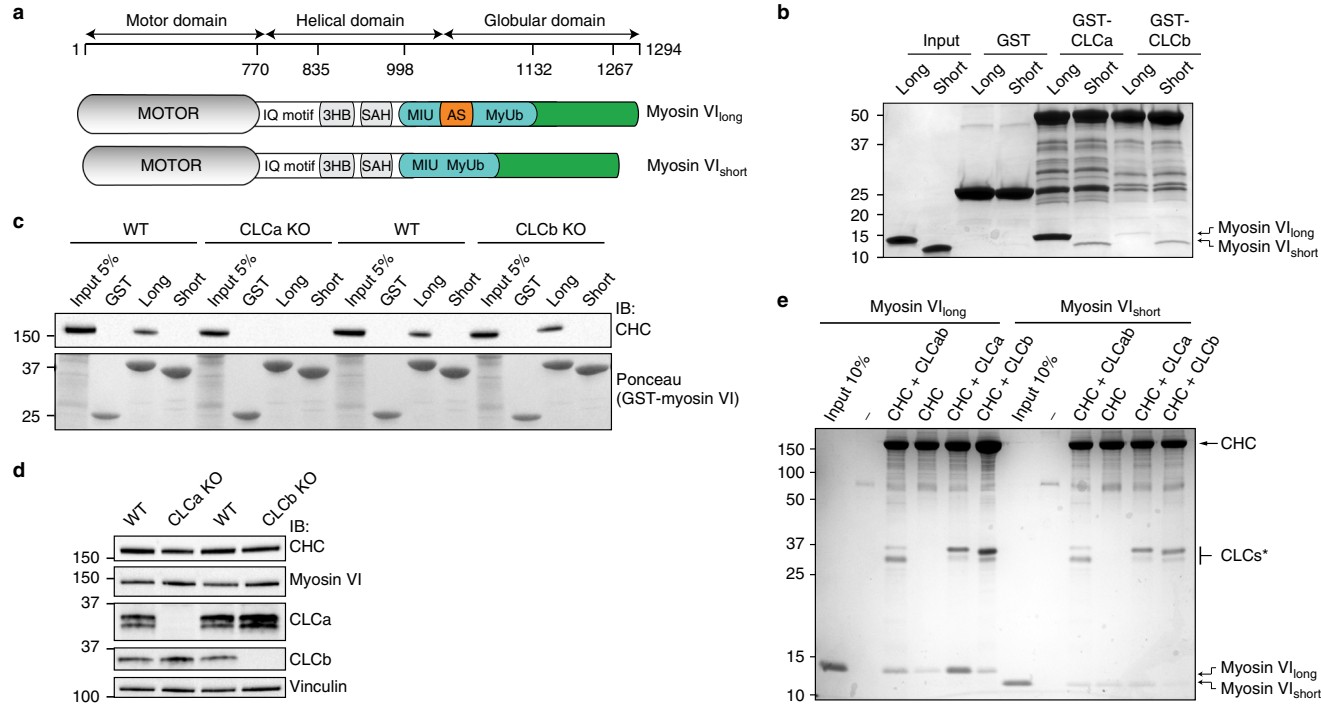

**Fig. 1** CLCa is a direct and specific interactor of myosin VI_long in triskelia and clathrin cages. **a** Scheme of the myosin VI highlighting the region involved in clathrin binding (amino acids 998–1131 of the long isoform). Long and short isoforms are reported together with the domains and motifs previously identified, including IQ motif, 3HB (three-helix bundle), SAH (single α-helix), MIU (motif interacting with ubiquitin), AS (alternative splicing region), and MyUb (myosin VI ubiquitin-binding domain). In orange is represented the alternatively spliced region codifying for the α2-linker[18]. **b** Pull-down assay with GST-CLCa and CLCb full-length and cleaved and purified fragments spanning amino acids 998–1131 of long and short myosin VI isoforms. Glutathione sepharose beads coupled to GST and GST-tagged proteins were incubated with myosin VI^998–1131. After washes, bound proteins were eluted in Laemmli-buffer, resolved through SDS-PAGE, and stained with Coomassie. **c** Pull-down assay using the long and short GST-myosin VI^998–1131 constructs and brain lysates (500 µg) obtained from the indicated mouse strains. After washes, bound proteins were eluted in Laemmli-buffer, resolved through SDS-PAGE, and transferred to a nitrocellulose membrane. Immunoblot (IB) was performed with anti-clathrin heavy-chain antibody. Ponceau detect equal loading of GST proteins. **d** IB of the brain lysates used in (**c**), as indicated. **e** Co-sedimentation assay. Equimolar (1.5 µM) amount of myosin VI^998–1131 and clathrin cages were incubated at 4 °C for 45 min in the presence of detergent (0.1% Triton X-100) and then pelleted by ultracentrifugation. Precipitated proteins were dissolved in Laemmli-buffer, resolved through SDS-PAGE, and stained with Coomassie. CLCs* indicates the various CLC proteins. Note that in the native cages CLCs (CHC-CLCab) run at different molecular weight (mw) as they are from pig brain while the human CLCs used for reconstitution are bacterially produced and cleaved from GST

paralogs (Fig. 2a), thus explaining the binding selectivity of CLCa to myosin VI_long. By analytical size-exclusion chromatography, the CLCa:47–97myosin VI^998–1131 sample co-eluted at the expected molecular weight for a 1:1 complex, indicating a stable and strong interaction in solution (Supplementary Fig. 2d). Further truncation analyses narrowed down the minimal interaction surface to a short peptide, with no secondary structure predicted, extending from amino acids. 51 to 61 (Fig. 2a and Supplementary Fig. 2e).

To evaluate the biochemical properties of the interacting CLCa peptide, we measured binding affinity of the myosin VI–CLCa interaction by fluorescent polarization (FP) analysis. We designed three different fluorescein-labeled CLCa peptides (Fig. 2a) that span the minimal binding surface (amino acids 51–61) exclusively or with five amino acids added at the N (amino acids 46–61) or C (amino acids 51–66) terminal end. FP measurements revealed a tenfold increase in affinity for the peptide 46–61 compared to the other two peptides (Fig. 2c), resulting in a dissociation constant ($K_d$) of 0.93 µM. Thus, in addition to the minimal binding region, amino acids spanning 46–51 conserved between CLCa and CLCb contribute to binding, which may explain the residual interaction observed by both CLCs to myosin VI_short (Fig. 1b).

We further tested the 46–61 peptide for binding to various myosin VI constructs to better define the interaction boundaries

on myosin VI. Confirming our previous results[18], the long and the short isoforms of myosin VI showed 2 log-fold difference in affinity for CLCa (0.93 µM versus 96 µM, respectively, Fig. 2c, d). As expected, the myosin VI^1069–1131 construct that lacks the α2-linker displayed a remarkably reduced $K_d$. Finally, the region encompassing amino acids 1050–1131 of myosin VI_long showed the highest affinity (0.4 µM, Fig. 2d) and was therefore chosen for the subsequent structural experiments.

### CLCa^46–61 forms a helix in a complex with myosin VI^1050–1131.
To further characterize the myosin VI:CLCa binding interface, we first compared two-dimensional (2D) ^1H, ^15N-HSQC spectra of myosin VI^1050–1131 before and after addition of the unlabeled CLCa peptide spanning 46–61 (Fig. 3a). Large effects were observed for many amino acids, which were quantified by chemical shift perturbation (CSP) analysis (Supplementary Fig. 3). Residues with CSP values of one standard deviation (SD) above average cluster to three regions in the myosin VI sequence (Supplementary Fig. 3) located at the N-terminal end of α2, N-terminal to α3, and toward the C-terminal end of α4. The CSPs above the mean were mapped onto the previously solved structure of free myosin VI (Fig. 3b), which revealed a localized region focused at the C-terminal half of α4 and the N-terminal region of α2.

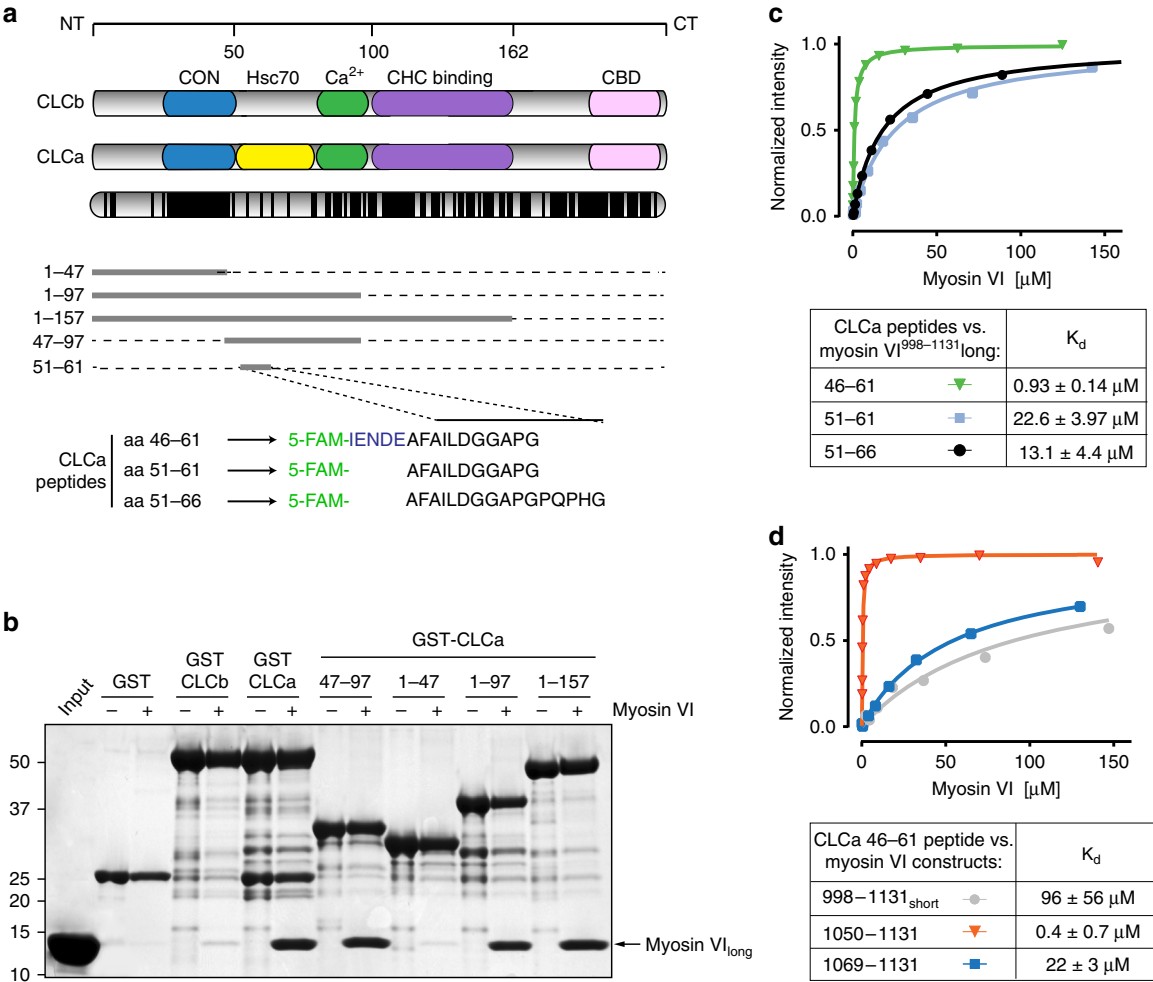

**Fig. 2** CLCa:myosin VI$_{long}$ interact with sub-micromolar affinity. **a** Domain structures of CLCa and CLCb. CON conserved Hip-binding region, Hsc70 unique region in CLCa that stimulates Hsc70 activity in vitro, Ca$^{2+}$ EF-hand domain that binds calcium, CHC binding clathrin heavy chain binding region, CBD calmodulin-binding domain. Sequence conservation between the two proteins is reported below. Each line represents one amino acid, black line indicates identity. Lower panel, scheme of the selected constructs used in (**b**) together with the sequence of the three overlapping 5-carboxyfluorescein (5-FAM)-conjugated CLCa peptides used for FP analysis in (**c**). **b** Pull-down assay with GST-CLCa and CLCb full length and the indicated fragments of CLCa immobilized on glutathione sepharose beads and incubated with the purified fragment spanning amino acids 998–1131 of myosin VI$_{long}$. After washes, bound proteins were eluted in Laemmli-buffer, resolved through SDS-PAGE, and stained with Coomassie. **c** FP assay using the three peptides shown in (**a**) and the purified fragment spanning amino acids 998–1131 of myosin VI$_{long}$. Dissociation constants with their respective 95% confidence interval (CI) are reported in the table at the bottom. Graph is representative of three independent experiments used to calculate $K_d$ and CI. **d** FP assay using peptide 46–61 of CLCa and the indicated fragments of long and short myosin VI isoforms. Graph, $K_d$, and CI as for (**c**)

In previously published structures of clathrin lattice and pits, the myosin VI-binding region of CLCa identified here was not detectable, as it falls into a flexible region[8,23]. As part of an effort to solve the structure of this CLCa region complexed with myosin VI$^{1050-1131}$ by NMR, we assigned chemical shift values to the myosin VI-bound CLCa fragment by 3D HNCACB (Supplementary Fig. 4) and nuclear Overhauser effect spectroscopy (NOESY; examples provided in Supplementary Fig. 5) experiments recorded on equimolar $^{15}$N, $^{13}$C-labeled CLCa and unlabeled myosin VI$^{1050-1131}$; we also assigned chemical shift values to CLCa-bound myosin VI, as described in Methods. The Cα and C' chemical shift assignments of CLCa were compared to those of randomly coiled values to generate a chemical shift index plot (Supplementary Fig. 6). This information was combined with NOE analyses obtained by NOESY experiments to define the secondary structure of CLCa (Supplementary Fig. 6) as being α-helical from residues I46–I54.

We next measured intermolecular interactions between myosin VI and CLCa directly by performing 3D $^1$H, $^{13}$C half-filtered

NOESY experiments; resulting spectra show NMR signals between intermolecular atoms that are within 5 Å of each other[24]. In agreement with the CSP analysis, we identified large numbers of intermolecular NOEs between CLCa and helices α2 and α4 of myosin VI (Supplementary Figs. 7 and 8). In particular, residues P1055, M1058, and M1062 from helix α2 (Supplementary Figs. 7a, b and 8) and V1120, Y1121, and W1124 from helix α4 (Supplementary Figs. 7c–f and 8) displayed numerous intermolecular NOEs to CLCa.

**CLCa binds to isoform-specific α2 and α4 of MyUb**. NMR data (summarized in Table 1), including 221 intermolecular NOE interactions between myosin VI and CLCa, were used to calculate the myosin VI:CLCa structure. The 20 lowest energy structures (Fig. 3c) converged with a backbone root mean square deviation (r.m.s.d.) in the ordered regions of 0.36 Å (Table 1). The myosin VI structure is largely unchanged in the complex, retaining its

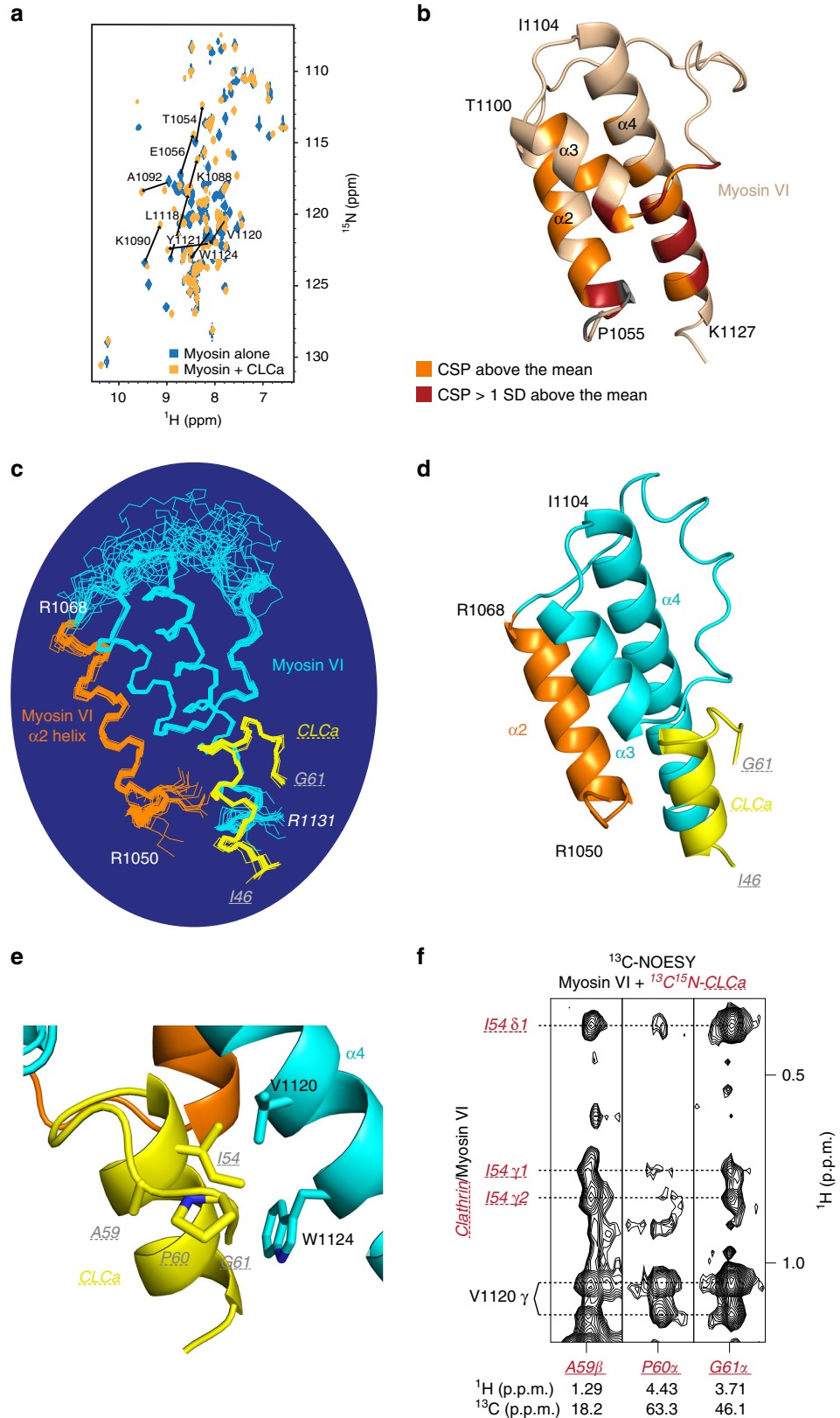

three helices and a disordered loop that follows the C-terminal end of isoform-specific α2. In agreement with our chemical shift index data (in Supplementary Fig. 6), CLCa$^{46-61}$ forms a short α-helix (yellow, Fig. 3c–e) that appears to complete the myosin VI structural domain by abutting the N-terminal end of α3 and packing against α2 and α4 (Fig. 3d). At the C-terminal end, CLCa L55-G58 forms a helical turn with A59-G61 folding back along one side of the CLCa α-helix (Fig. 3e), a structure supported by NOEs to CLCa I54 and to myosin VI α4 (V1120 and W1124, Fig. 3f and Supplementary Fig. 7c, e). The hydrophobicity of CLCa A59 and P60 likely drives this segment to remain in close contact with the helices (Fig. 3e), as supported by the NMR data (Fig. 3f).

**Fig. 3** The myosin VI-binding region of CLCa forms an α-helix that interlocks with the helices of myosin VI. **a** 2D $^1$H, $^{15}$N-HSQC spectrum of 0.78 mM $^{15}$N-labeled myosin VI$^{1050-1131}$ alone (blue) and in the presence of equimolar unlabeled CLCa$^{46-61}$ (orange). Notable changes in NMR signals following addition of CLCa are indicated with arrows and labeled with the assigned amino acid residue from myosin VI. **b** On the previously solved structure of myosin VI$^{1050-1131}$ (PDB 2N12), amino acids with CSPs greater than the mean or 1 SD above the mean are indicated in orange or red, respectively. CSPs were calculated according to the definition CSP = $[(0.2\Delta\delta N)^2 + (\Delta\delta H)^2]^{1/2}$ where $\Delta\delta N$ and $\Delta\delta H$ represent chemical shift differences for the amide nitrogen and proton of each residue, respectively. Residues without amide protons (P1051, P1055, and P1070) are indicated in gray, and those at the beginning and end of each helix are labeled. **c** Superimposition of the 20 lowest energy structures calculated for myosin VI$^{1050-1131}$ in complex with CLCa$^{46-61}$. CLCa is shown in yellow; myosin VI is colored cyan except for the isoform-specific α2 helix, which is highlighted in orange. The orientation of the structures is identical to **b**. **d** Ribbon representation of the structure of myosin VI bound to CLCa$^{46-61}$ using the same color scheme and orientation as in **c**. **e** Ribbon representation as in **d**, expanded and rotated about the CLCa helix to highlight the α-helical structure of CLCa$^{46-54}$ and interactions involving the C-terminal amino acids of the CLCa peptide. Sidechain heavy atoms of the highlighted residues are included. Note that the C-terminal portion of the CLCa fragment remains in close contact with myosin VI and the CLCa helix. **f** Selected regions from a 3D $^{13}$C NOESY experiment acquired on 0.4 mM $^{13}$C, $^{15}$N-labeled CLCa$^{46-61}$ and equimolar unlabeled myosin VI$^{1050-1131}$. NOE interactions involving the C-terminal portion of CLCa (A59, P60, and G61) are shown. Intramolecular and intermolecular interactions are labeled in red and black, respectively

**Table 1 NMR data collection and refinement statistics**

|  | Total | Myosin VI$^{1050-1131}$ | CLCa$^{46-61}$ |
|---|---|---|---|
| NMR distance and dihedral constraints |  |  |  |
| NOE-derived distance constraints |  |  |  |
| Total NOE | 2540 |  |  |
| Intramolecular |  | 1971 | 348 |
| Intra-residue |  | 687 | 91 |
| Inter-residue |  | 1284 | 257 |
| Sequential ($\|i - j\| = 1$) |  | 509 | 105 |
| Medium-range ($2 < \|i - j\| < 4$) |  | 545 | 126 |
| Long-range ($\|i - j\| > 5$) |  | 230 | 26 |
| Intermolecular | 221 |  |  |
| Hydrogen bonds |  |  |  |
| Total | 60 |  |  |
| Intramolecular |  | 47 | 9 |
| Intermolecular | 4 |  |  |
| Total dihedral angle constraints |  |  |  |
| $\phi$ |  | 50 | 9 |
| $\psi$ |  | 50 | 9 |
| Structure statistics |  |  |  |
| Violations (mean and s.d.) |  |  |  |
| Distance constraints (Å) | 0 |  |  |
| Dihedral angle constraints (°) | 0 |  |  |
| Max. dihedral angle violation (°) | <5 |  |  |
| Max. distance constraint violation (Å) | <0.3 |  |  |
| Deviations from idealized geometry |  |  |  |
| Bond lengths (Å) | 0.003 ± 0.001 |  |  |
| Bond angles (°) | 0.420 ± 0.013 |  |  |
| Impropers (°) | 0.302 ± 0.009 |  |  |
| Average pairwise r.m.s. deviation$^a$ (Å) |  |  |  |
| Heavy | 0.99 ± 0.14 |  |  |
| Backbone | 0.36 ± 0.09 |  |  |

$^a$Values were calculated for the 20 lowest energy structures of myosin VI residues M1053–R1068 and L1086–N1128 and CLCa residues E47–P60

As predicted by the binding selectivity of the long isoform[18], the interaction surface on myosin VI involves isoform-specific residues from α2, including P1055, M1058, A1059, and M1062, which surround L55 of CLCa (Fig. 4a, b). Myosin VI P1055 extends this CLCa-binding surface through its interaction with F52 (Fig. 4b, NOEs shown in Supplementary Fig. 7b). At an adjacent side of the CLCa helix, myosin VI R1117, V1120, Y1121, and W1124 surround CLCa A51 and I54 (Fig. 4a, b). In addition, the W1124 indole group forms a hydrogen bond to the sidechain carboxyl group of E50 (Fig. 4a).

The RRL motif required for binding to multiple adaptor proteins including optineurin and GIPC[18,25,26] is embedded in α4 (Supplementary Fig. 9a). R1116 does not participate in the interaction and remains surface exposed, whereas both R1117 and

L1118 contribute to CLCa binding. R1117, required for myosin VI structural integrity[18,27], maintains its hydrogen bonds to S1087 and E1113 as in free myosin VI and forms a hydrogen bond to the backbone oxygen of CLCa D56, the sidechain of which also forms a hydrogen bond to the backbone amide of myosin VI Y1091 (Fig. 4c). Lastly, L1118 of the RRL motif contributes to binding though interactions with CLCa L55 (Fig. 4b). Notably, the CLCa amino acids critical for binding to myosin VI, including A51, I54, L55, and D56, are not conserved in CLCb (Fig. 4d), thus providing an explanation for paralog specificity.

The importance of the identified interactions is supported by GST pull-down experiments. A truncated construct confirmed that the α4 helix of myosin VI is involved in binding to CLCa

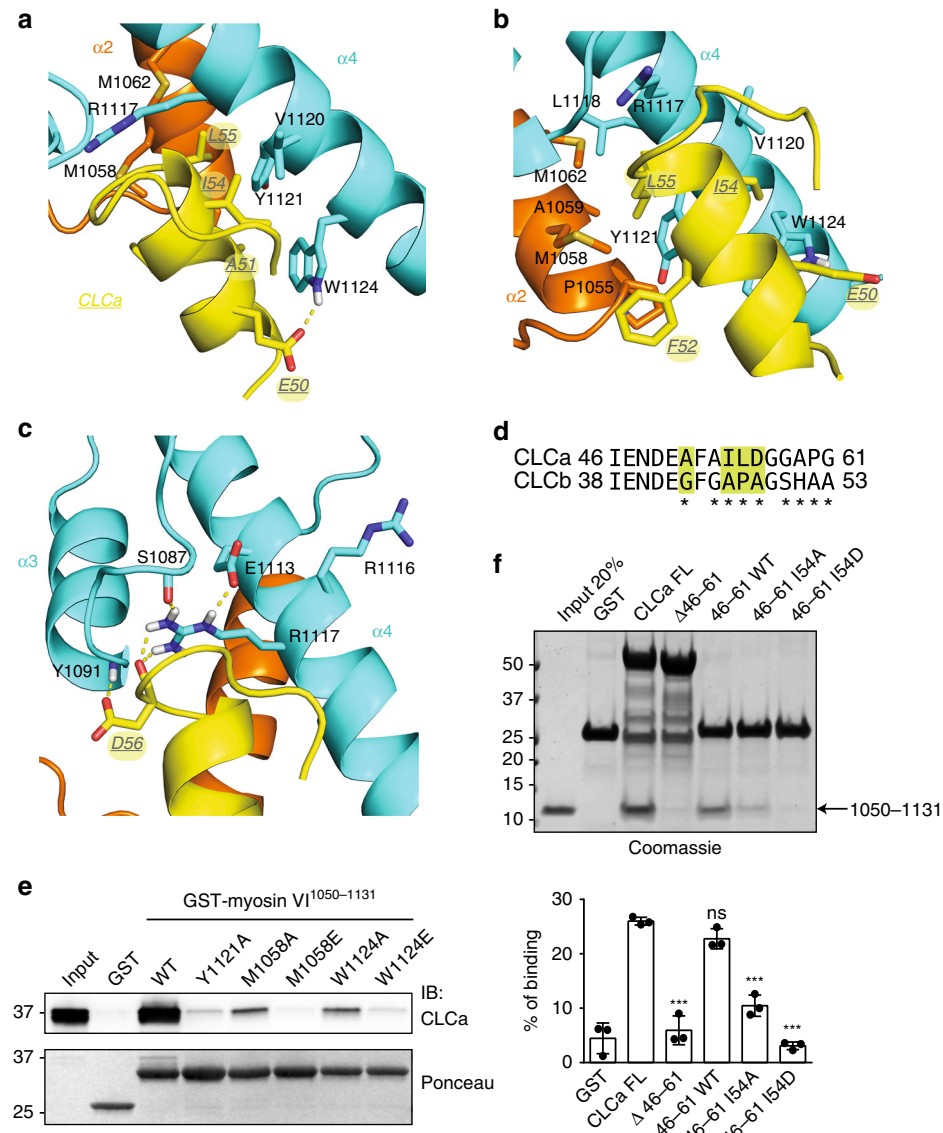

**Fig. 4** A hydrophobic pocket formed by myosin VI encompasses residues I54 and L55 of CLCa. **a** Ribbon representation of the myosin VI:CLCa binding interface in the orientation of Fig. 3e with key sidechain heavy atoms displayed. Myosin VI α4 residues R1117, V1120, Y1121, and W1124 (blue) interact with CLCa E50, A51, I54, and L55 (yellow). CLCa L55 also interacts with myosin VI α2 residues, with M1058 and M1062 (orange) observable in this view. At the edge of the hydrophobic pocket lies a hydrogen bond between CLCa E50 and the myosin VI indole group. Sulfur and nitrogen atoms are in yellow and indigo, respectively. **b** As in **a** but rotated about the CLCa helix to highlight interactions involving the myosin VI isoform-specific α2 helix, especially P1055 and A1059. **c** Enlarged representation of the region containing myosin VI R1117 to highlight intramolecular and intermolecular hydrogen bonds. The guanidine group of R1117 forms hydrogen bonds to the backbone and sidechain carboxyl groups of myosin VI S1087 and E1113, respectively, and to the backbone carboxyl group of CLCa D56. This view is similar to that of **a** but rotated about the sidechain of R1117. In **a**, **c**, a dashed yellow line is used to indicate a hydrogen bond with oxygen and nitrogen atoms in red and indigo, respectively. **d** Sequence alignment of CLCa 46–61 with the corresponding region of CLCb. Asterisks indicate residues not conserved between isoforms. In yellow are amino acids putatively responsible for the selective binding. **e** Pull-down assay using the indicated GST-myosin VI$^{1050-1131}$ mutant constructs and lysates (1 mg) from HEK293T cells. After washes, bound proteins were eluted in Laemmli-buffer, resolved through SDS-PAGE, and IB was performed with the anti-CLCa antibody (×16). Ponceau detects equal loading of GST proteins. Representative image of three independent experiments is shown. **f** GST pull-down assay using the indicated CLCa constructs and purified fragment spanning amino acids 1050–1131 of myosin VI$_{long}$. After washes, bound proteins were eluted in Laemmli-buffer, resolved through SDS-PAGE, and stained with Coomassie. Bottom panel, quantitation of three independent experiments. Data are expressed as percentage of binding with respect to input and normalized for the amount of GST proteins used in each pull-down. Error bars represent s.d. ***$P < 0.001$ by two-tailed $T$ test

(Supplementary Fig. 9a) while single substitution of myosin VI M1058, Y1121, or W1124 led to reduced binding (Fig. 4e). FP analysis revealed a 2 log-fold difference in binding affinity for the Y1121A mutant (Supplementary Fig. 9b). On the CLCa side, we tested the effect of substituting I54 with alanine or aspartic acid, using CLCa full-length protein as a control. As expected, myosin VI$^{1050-1131}$ bound to CLCa WT but not Δ46-61. I54A or I54D

significantly impairs binding to myosin VI, with aspartic acid showing the strongest defect (Fig. 4f).

**Myosin VI requirement for CME in polarized cysts**. While CLCa is ubiquitously expressed in animal tissues[5], the presence of myosin VI$_{long}$ is restricted to organs containing polarized cells of

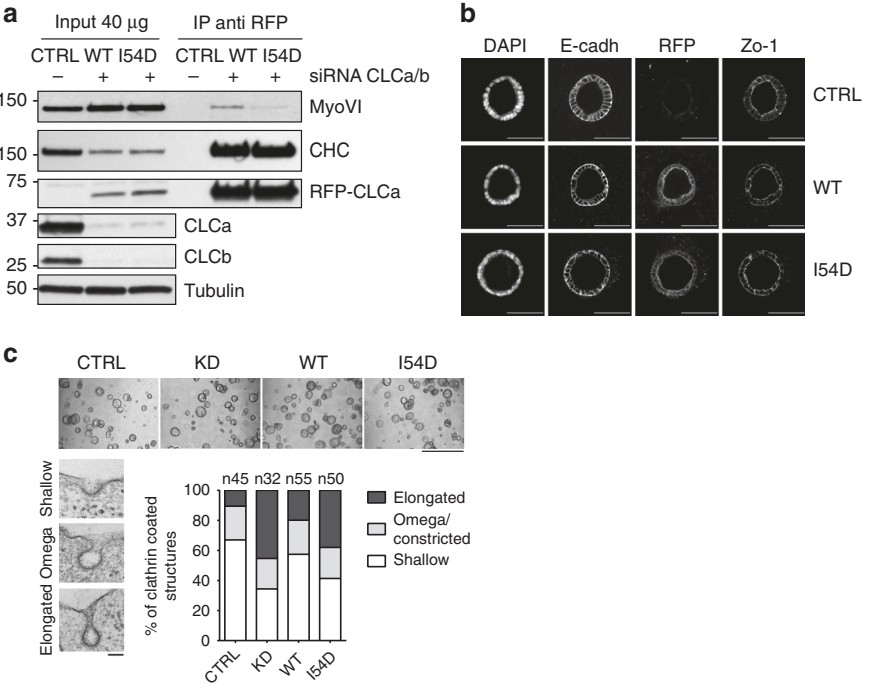

**Fig. 5** CLCa I54D is a selective myosin VI-impaired mutant. **a** Co-immunoprecipitation experiment using lysates from Caco-2 reconstituted cell lines. Cells depleted of endogenous CLCs or mock treated are kept in confluency for 7 days, lysed, and RFP-CLCa wild-type and I54D were immunoprecipitated from 1 mg of lysates. IP and IB as indicated. **b** Representative confocal micrographs of CLCa wild-type and I54D localization in Caco-2 cysts. Staining with the indicated apical–basal polarity markers are shown. Scale bar, 100 μm. **c** Upper panel, representative brightfield pictures of Caco-2 cysts generated using the indicated isogenic cell lines. Scale bar, 1 mm. Clathrin-coated structures present at the apical surface of the Caco-2 cysts were counted and classified as shallow, omega/constricted, or elongated according to their morphology. Left panel, representative EM images of the different morphology of CCPs. Scale bar, 100 nm. Right panel, distribution of clathrin-coated structures in the different isogenic cell lines expressed as percentage among the different classes and calculated on three independent experiments. Between 94 and 125 cellular profiles of well-polarized cells for each line were imaged. Number of clathrin-coated structures counted in total are reported as $n = x$. See also Supplementary Fig. 11d

epithelial origin, such as kidney and intestines, both in mice[28] and humans (Supplementary Fig. 10a). There, myosin VI localizes to the apical surface facing the lumen of the organs at the base of microvilli[29,30] (Supplementary Fig. 10b). To analyze the physiological role of the CLCa:myosin VI complex in a cellular model of polarized epithelial tissue, we took advantage of the intestine-derived epithelial Caco-2 cells that form polarized cysts when plated as a single-cell suspension embedded in 3D EHS-derived matrix[31]. Notably, in this Caco-2 cellular model system, a clear switch toward the myosin VI$_{long}$ isoform occurs during the acquisition of full polarity both in 2D and 3D systems, as measured by reverse transcriptase–polymerase chain reaction (PCR) (Supplementary Fig. 10c). Transmission electron microscopy (TEM) and confocal microscopy analysis showed that the cysts are fully formed and polarized (Supplementary Fig. 10d–f) and myosin VI$_{long}$ is enriched in the apical terminal web region together with occludin (Supplementary Fig. 10d). We then generated Caco-2 cells stably expressing red fluorescent protein (RFP)-WT or an RFP-I54D mutant rat CLCa as these constructs are resistant to the small interfering RNA (siRNA) oligos designed on the human sequence. Upon efficient depletion of the endogenous CLCa and CLCb by siRNA oligos (Supplementary Fig. 11a–c and Fig. 5a), co-immunoprecipitation analysis performed with lysates from 2D fully polarized Caco-2 cells demonstrated that the I54D mutant was largely unable to interact with myosin VI (Fig. 5a), validating our previous in vitro results. Next, single Caco-2 reconstituted cells depleted of endogenous CLCs were cultured in matrigel and 7 days after cysts were counted and stained to evaluate adherens and tight junctions and the localization of RFP-CLCa WT or I54D mutant (Fig. 5b).

Exogenous RFP-CLCa I54D mutant, similarly to the WT protein, was enriched in the apical region toward the lumen and did not appear to affect the ability of the cells to form cysts (Fig. 5c) or prevent the establishment of apical basal polarity (Fig. 5b and Supplementary Fig. 10e, f).

Clues as to the molecular requirement for CLCa–myosin VI interaction were derived from EM analysis of the cysts. As CLCs are implicated in CME, we investigated the CCPs that formed at the apical surface facing the lumen in control, CLC-depleted, and reconstituted RFP-CLCa variants. CCPs stemming from the PM were classified as shallow, omega/constricted, or elongated according to their morphology (Fig. 5c, left panel). Compared to mock-treated Caco-2 cysts, cysts depleted of CLCs displayed a significantly greater proportion of CCPs with an aberrant morphology, consisting of elongated pits joined to the PM by "long necks" (Fig. 5c). These structures closely resembled those induced by the dominant-negative mutant of dynamin, DynK44A[32]. Strikingly, while WT RFP-CLCa fully rescued this phenotype, the I54D mutant was unable to do the same, indicating that the morphological defect scored in the absence of CLCs can be ascribed to the lack of myosin VI interaction with CLCa (Fig. 5c, Supplementary Fig. 11d).

These results are compatible with the possibility that the interaction of CLCa with myosin VI is part of the machinery that controls CCP fission. Previously, CLCs and their binding partner Hip1R were implicated in this process at the apical membrane of polarized cells[6]. Intriguingly, the myosin VI-binding region we identified here on CLCa partially overlaps with the Hip1R-binding region[7] (Fig. 6a). Thus we evaluated the ability of myosin VI and Hip1R to bind simultaneously to CLCa, using an in vitro

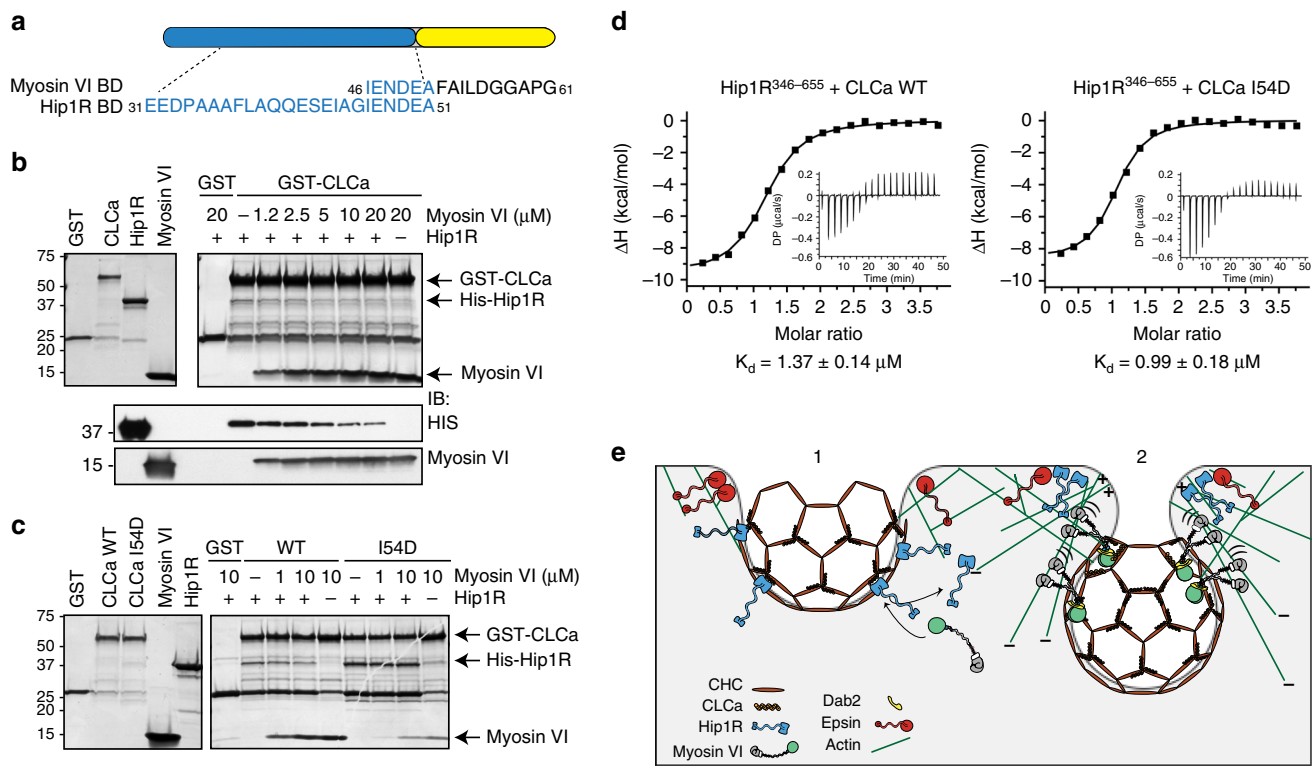

**Fig. 6** Myosin VI and Hip1R are mutually exclusive binders of CLCa. **a** Binding regions of myosin VI and Hip1R on CLCa. **b** Competitive binding assay. Bacterially purified His-Hip1R coiled-coil region spanning residues 346–655 (2.5 μM) pre-incubated with GST-CLCa full-length protein (1 μM) for 1 h at 4 °C was mixed with increasing amounts of myosin VI[1050-1131] as indicated. Bottom panel, Coomassie staining. Lower panels, 1/10 of the elutant was loaded for immunoblotting with the indicated antibodies. **c** As in **b** but using GST-CLCa I54D mutant. **d** ITC experiments with the indicated proteins. The integrated heat and raw plots are reported. Equilibrium dissociation constants ($K_d$) obtained by the fitting are indicated below. Relevant ITC measurements are reported in Table 2. **e** A putative model of clathrin-coated pit fission in polarized tissue. Once CCP passes the endocytic checkpoint, it undergoes a maturation process mediated by PIP2 turnover and the activity of several endocytic accessory factors (e.g., Hip1R). (1) As the bud expands, myosin VI is recruited to CCPs by CLCa and disengages Hip1R, which preferentially associates with epsin at the CCP edge. Actin anchoring and polymerization is thereby restricted to the neck of the invaginating pit where Hip1R along with epsin provide a link between actin nucleation and the CCP. Alternatively, formation of a complex between epsins and Hip1R at the neck would recruit Hip1R from the coat, exposing the myosin VI-binding site on CLCa and causing recruitment of myosin VI to the CCP. (2) Myosin monomer bound to CLCa anchors the CCP to the actin meshwork engaged in retrograde flow and facilitates movement of the vesicle into the cytoplasm. After dimerization induced by Dab2 or oligomerization, myosin VI becomes a processive motor that walk toward the minus end of actin filaments providing mechanical force to oppose membrane tension in polarized tissues and promoting the final dynamin-mediated fission step (not depicted). Future experiments are needed to test this model. In the myosin VI representation, the boxing glove and white cuff indicate the motor domain and single IQ motif, respectively, while the cargo-binding tail, which includes the clathrin-binding domain, is depicted as a green sphere; the black squiggly connecting region includes the 3HB and SAH

**Table 2 Thermodynamic parameters obtained by isothermal titration calorimetry**

|  | *N* (sites) | Δ*H* (kcal/mol) | Offset (kcal/mol) | Δ*G* (kcal/mol) | −*T*Δ*S* (kcal/mol) | $K_d$ (μM) |
|---|---|---|---|---|---|---|
| Hip1R + CLCa FL | 1.15 + 0.01 | −9.70 + 0.19 | 2.27 + 0.06 | −8.00 | 1.69 | 1.37 + 0.14 |
| Hip1R + CLCa I54D | 1.01 + 0.02 | −8.72 + 0.27 | 1.25 + 0.08 | −8.19 | 0.528 | 0.99 + 0.18 |
| Myo6 + CLCa FL | 0.79 + 0.02 | −11.6 + 0.47 | 2.02 + 0.18 | −9.00 | 1.69 | 0.09 + 0.05 |
| Myo6 + CLCa I54D | N/A | N/A | N/A | N/A | N/A | N/A |
| *N/A* not applicable |  |  |  |  |  |  |

assay with bacterially purified components. We incubated GST-tagged CLCa full-length protein with the coiled-coil region of Hip1R spanning amino acids 346–655 and added increasing amounts of myosin VI[1050–1131] protein fragment. Analysis by Coomassie staining and immunoblotting showed that myosin VI addition caused progressive displacement of Hip1R from CLCa (Fig. 6b), indicating a direct competition between the two CLCa interactors possibly due to steric hindrance. Consistent with separate but proximal binding sites for these two CLCa-binding

proteins, Hip1R bound both the CLCa I54D mutant and WT CLCa, but its binding to the mutant was not affected by myosin VI (Fig. 6c). Notably, CLCb interaction with Hip1R was insensitive to the addition of myosin VI in the same competition assay (Supplementary Fig. 11e), confirming once again the selectivity of myosin VI for CLCa. Finally, we measured the affinity of the two interactors for CLCa full length by isothermal calorimetric assay (isothermal titration calorimetry (ITC), Table 2). Myosin VI bound CLCa with a $K_d$ of 90 nM, while it

showed no binding to CLCa I54D (Supplementary Fig. 11f). In agreement with the competition assay, Hip1R binds WT and mutant CLCa equally well but with a $K_d$ a log higher with respect to myosin VI (Fig. 6d).

Altogether, these data provide evidence for coordinated action of myosin VI and CLCa in the fission of CCPs at the apical surface of polarized cells and identify a direct competition between Hip1R and myosin VI.

## Discussion

It is well established that actin remodeling during CME is required when the PM tension is high[6] or when there is a need to change the mechanical properties of the clathrin lattice to engulf non-canonical bulky cargo, such as viral or bacterial pathogens[10,12]. The functional link between CCPs and actin is represented by the CLC–Hip1R interaction in both yeast and mammals[33–35]. Here we identified the role of a third player in this process, myosin VI, which selectively binds to CLCa and directly competes with Hip1R. While myosin VI has been previously implicated in CME[26,36,37], our structural definition of the CLCa–myosin VI interaction provides the molecular details of such involvement.

Myosin VI is unique among mammalian myosins in that it moves toward the minus end of F-actin[38]. This unique motor protein supports critical functions at multiple cellular sites via interaction with different adaptors that target it to specific organelles. We recently showed that an alternatively spliced α2-linker modulates the myosin VI interactome and is the causative link to its diverse functions[18]. In particular, the myosin VI$_{long}$ isoform includes the exon cassette that codes for the α2-linker and is endowed with clathrin-binding capacities and endocytic functions[18].

Here we reveal that myosin VI is recruited to CCPs via a selective interaction with CLCa. Our biochemical and structural characterization of the myosin VI–CLCa interaction identified contacts unique to CLCa that are contributed largely from a helical structure. Key differences in the amino acid sequence of the CLCa and CLCb paralogs are likely responsible for the lack of binding between myosin VI and CLCb (Fig. 4d). Specifically, A51 and I54 in CLCa are both replaced with smaller residues (Gly and Ala, respectively, in CLCb), which likely diminishes the hydrophobic contacts with myosin VI (Fig. 4a, b). In CLCb, D56 is replaced with an alanine, which cannot form a hydrogen bond with Y1091 in myosin VI (Fig. 4c). Finally, L55, which embeds itself between helices α2 and α4 of myosin VI (Fig. 4a, b), is replaced with a proline in CLCb. Owing to the structural rigidity of proline, this amino acid may disrupt the helical turn, possibly preventing CLCb from fitting into the myosin VI-binding pocket that accommodates CLCa.

The CLCa peptide identified here as determinant for myosin VI interaction was invisible in the previously published structures of the clathrin lattice and the central domain of the triskelion[8,23] and identified as a conformationally labile peptide modulated by calcium ion concentration[39], strongly suggesting that this flexible peptide is an important element of regulation for CLC. Calcium was also shown to mobilize myosin VI from a compact dormant state into a cargo-binding primed state[40]. Interestingly, the segment of the myosin VI tail region mediating this regulation (amino acids 1060–1125) matches exactly with the clathrin-binding domain. In resting conditions, myosin VI adopts a backfolded, dormant state, with the tail domain amino acids 1060–1125 binding to apo-calmodulin; at high calcium levels, myosin VI unfolds and the release of the myosin VI tail mobilizes the motor, priming it to bind to cargo[40]. In this open structural conformation of myosin VI, CLCa binding should be more

favorable. Consistently, addition of 2 mM $Ca^{2+}$ to the lysis buffer significantly improved the co-immunoprecipitation between myosin VI and CLCa (Supplementary Fig. 12). Thus our results, together with these previous findings, suggest that release of intracellular calcium could orchestrate the timing of localized myosin VI activation and of clathrin–myosin VI functional interaction.

Like Hip1R[41] and CLC[42], myosin VI is largely dispensable for CME in 2D culture in vitro. However, the specific expression of myosin VI$_{long}$ in polarized tissues (e.g., kidney and intestine), together with its features as an actin-based motor, prompted us to examine its function in Caco-2 cysts, used as proxy of the enterocytes lining the small intestine.

We show here that, in Caco-2 grown as cysts, a clear switch toward myosin VI$_{long}$ occurs as cysts develop with this endogenous isoform enriched at the apical membrane, together with clathrin-rich structures. As further discussed below, CME in epithelial polarized cells is dependent on actin polymerization, which drives membrane invagination against the membrane tension that rigidifies the apical membrane[6]. In this process, CLCa may recruit myosin VI to CCPs to provide the force necessary to complete neck constriction, fission, and release. Indeed, using a genetic reconstitution approach, we were able to demonstrate a clear change in the distribution of CCP structures at the apical surface when we abrogated the CLCa–myosin VI interaction. The increased number of elongated CCPs present at the apical surface of CLC-depleted cysts or cysts reconstituted with I54D CLCa mutant, which does not bind myosin VI, suggests a delay in CCP progression at this surface. Our results are compatible with the idea of myosin VI acting at the neck of a mature CCP to promote fission. Consistent with this notion, recent evidence demonstrated that myosin VI, together with branched actin filaments, mediates severing by constricting the neck of budding tubules that are released from melanosomes[43].

The involvement of the myosin VI–CLCa interaction in the CCP fission step at the apical membrane of polarized cells is attractive, as this step is where actin has also been implicated in yeast[44] and observed in mammalian cells[45]. It is tempting to speculate that the lack of CLCa–myosin VI interaction and thus a defective CME is the cause of the altered level of CTFR, megalin receptor and sodium type-IIa cotransporter (NaPi2a) observed at the apical membrane of myosin VI-depleted enterocytes and renal proximal tubular cells[19,46–51].

Our results unveil a direct competition between myosin VI and Hip1R for CLCa (Fig. 6a–d) that likely establishes partitioning of actin activity during the various stages of CCP formation (Fig. 6e). Current knowledge localizes Hip1R bound to CLCs throughout the CCP as well as inserted directly in the surrounding PM via its lipid-binding ANTH domain[52,53]. The conformational change induced by CLC binding masks the actin-binding domain of Hip proteins in the coat[8] and would exclude actin interaction at this site. At the adjacent membrane, Hip protein interaction with the epsin family proteins, both in yeast[52] and in mammals[54], results in a multimeric heterocomplex[52,55] that may preferentially occur at the neck of an invaginating CCP[56]. Here Hip proteins dissociated from the clathrin coat could interact with actin, generating the active site required for the invagination of coated pits. Indeed, depletion of either Hip1R or epsins results in similar abnormal actin nucleation at endocytic sites where vesicle budding is inhibited[33,54]. Thus regulation of actin activity by Hip proteins seems to be restricted to the neck of an invaginating vesicle[8] where barbed ends of branched actin filaments are concentrated[45] and fission occurs as dynamin assembles into a helical coat[57,58]. How might this happen? We propose here that myosin VI contributes to the temporal regulation of Hip1R action while producing the forces to overcome high membrane tension (Fig. 6e). During

vesicle invagination, myosin VI recruited by CLCa would compete out Hip1R, promoting its dissociation from the cages and recruitment by epsins at the uncoated neck. At the vesicle neck, Hip1R regulates actin filament polymerization[33] while Dab2 may tether myosin VI recruited by CLCa into a stable and active dimer[59]. Finally, acting as a processive cellular motor on branched actin filaments, myosin VI may contribute to the release of the invaginating vesicle, promoting CCP fission (Fig. 6e). Notably, our findings are in line with previous studies based on total internal reflection fluorescence microscopy that revealed myosin VI to be recruited during a late stage of clathrin-mediated internalization, concomitantly with dynamin and regulators of actin polymerization, to promote the scission of the invaginated membrane[60]. Consistent with the idea of myosin VI actively participating in the latest steps of CCP maturation, in vivo analysis of renal proximal tubule cells in mice lacking myosin VI showed reduced uptake of endocytic markers and accumulation of AP2 and Dab2 at the PM[49].

Our findings indicate that only CLCa-containing CCPs may be subjected to the regulation exerted by myosin VI. CLCa-null mice lack the myosin VI–CLCa interaction and its unique function and this deficiency may contribute to the 50% postnatal mortality rate of these animals[5]. Consistent with this idea, CLCa is ubiquitously expressed at high levels in all vertebrate tissues, whereas CLCb characteristically exhibits variable levels[5], and compared to CLCb, CLCa is preferentially associated with peripheral clathrin-coated structures and proteins involved in cell spreading[15], a process that also requires actin regulation.

We are at an early stage of appreciating the extent of the selective activity of each CLC paralog, with hints of CLCa- and CLCb-associated functions emerging in recent reports[15–17]. The high conservation and distinction of the two CLC-encoding genes that are preserved within all vertebrates[61] strongly support the notion that CLCa and CLCb proteins have non-redundant functions. In the emerging scenario, our data indicate CLCa as a key modulator of clathrin interactions with the actin cytoskeleton.

## Methods

**Cell lines and KO mouse strains and constructs.** HEK293T cells (ICLC) were grown in Dulbecco's Modified Eagle Medium (DMEM) supplemented with 10% fetal bovine serum (FBS) S.A. and 2 mM L-glutamine. HeLa cells (ATCC) were grown in Minimum Essential Medium (MEM) supplemented with 10% FBS S.A., 0.1 mM non-essential amino acids, 2 mM L-glutamine, and 1 mM Na-Pyruvate. CaCo-2 cells (ATCC) were grown in MEM supplemented with 20% FBS S.A., 0.1 mM non-essential amino acids, and 2 mM L-glutamine.

At each batch of freezing, all cell lines were authenticated by STR profiling (StemElite ID System, Promega) and tested for mycoplasma using PCR and biochemical test (MycoAlert, Lonza).

Caco-2 RFP-CLCa WT and CLCa I54D isogenic cell lines were generated by lentiviral infection of Caco-2 cells with the corresponding pLVX-RFP expressing rat-CLCa variants and bulk populations were FACS-sorted to enrich for RFP-positive cells.

Production of mice deleted for the *CLTA* gene encoding CLCa (CLCa KO) was previously described[5]. Mice deleted for the *CLTB* gene encoding CLCb (CLCb KO) were generated in the Brodsky laboratory and loss of CLCb in the tissues studied here has been confirmed, as verified in Supplementary Fig. 2a.

Human constructs of myosin VI were previously described[18,27]. Mutants generated here were engineered by site-directed mutagenesis and sequence verified. Details are available in Supplementary Table 1.

pRSF-duet codifying for His-Hip1R coiled coil region (amino acids 346–655) was previously described[8].

pGEX6P1-CLCa FL construct was generated by subcloning RFP-hCLCa, kindly provided by Klemens Rottner (HZI, Braunschweig), into pGEX6P1 with XhoI enzyme. Subsequent truncated constructs were engineered by site-directed mutagenesis or recombinant PCR and sequence verified. Details are available upon request.

pGEX6P1-CLCb FL construct was generated by PCR using as template pCMV tag 2B-hCLCb, kindly provided by Elizabeth Smythe (University of Sheffield), and sequence verified.

pLVX lentiviral vector expressing rat RFP-CLCa was cloned from pMSCV-RFP-ratCLCa vector, kindly provided by Tom Kirchhausen (Harvard Medical School, Boston), using the InFusion system (Takara Clontech) and the following oligos:

forward: CTCAAGCTTCGAATTCATGGTGTCTAAGGGCGAAGAGCTG
reverse: GTCGACTGCAGAATTCTCAATGCACCAGGGGCGC.

pLVX RFP-CLCa I54D mutation was generated by site-directed mutagenesis and sequence verified.

pEGFP-CHC and pGEX6P1-CHC constructs described in Supplementary Fig. 1a were kindly provided by Stephen Royle (University of Liverpool).

5-Carboxyfluorescein (5-FAM)-conjugated CLCa peptides 46–61, 51–61 and 51–66 were synthesized by GenScript.

**Antibodies.** All antibodies used in this study are listed in Supplementary Table 2.

**CLC depletion.** Transient CLCa and/or CLCb KD were performed using Stealth siRNA oligonucleotides from ThermoFisher Scientific (humanCLCA-3utr target sequence: 5′-TGGAAACACTACATCTGCAATATCT-3′ and humanCLCB-3utr target sequence: 5′-CGCCTCCTCTCAGTCTACTCAATTG-3′) or Qiagen [humanCLCa (2) target sequence: AGACAGTTATGCAGCTATT].

Caco-2 (or HeLa, Supplementary Figs. 1 and 11d) cells were transfected using RNAiMax (Invitrogen) and siRNA oligos at 8 nM final concentration. Cells were transfected twice, first in suspension and the day after in adhesion. The third day Caco-2 cells reach confluency and were kept confluent for additional 7 days in order to obtain a compact polarized monolayer. Medium was changed every 2 days.

**Caco-2 cyst formation.** Caco-2, Caco-2 RFP-rCLCa WT, and Caco-2 RFP-rCLCa I54D cell lines were transiently transfected with CLC oligos. Two days after siRNA transfection, single-cell suspensions were embedded in medium plus 50% Matrigel (Corning) and 1 mg/ml collagen mixture and plated on 8-well ibidi chamber (Cat. No: 80826) at a density of $9 \times 10^3$ cells/well. Approximately 150 μl were plated in each well, allowed to solidify for 30 min at 37 °C, and overlayed with 200 μl of medium. Medium was changed every 2 days. At day 6, cholera toxin (Sigma Cat. No.: C8052) was added at a concentration of 0.1 μg/ml and cysts are collected the day after.

**Immunofluorescence (IF) and co-localization experiments.** For IF analysis, samples were fixed with 4% paraformaldehyde for 10 min and permeabilized at room temperature (RT) with 0.5% Triton-X100 in PBS for 10 min (cells) or 20 min (mice tissue). Blocking was performed in phosphate-buffered saline (PBS) containing 2% bovine serum albumin (BSA) for 1 h prior to the incubation of the primary antibodies in PBS containing 0.1% Triton X-100 and 2% BSA overnight at 4 °C. After extensive washing, sample were incubated with secondary antibodies 45 min, at RT. Coverslips were mounted in a glycerol solution (20% glycerol, 50 mM Tris pH = 8.4) to avoid mechanical deformation of the sample. Images were captured using a HCX ×63 plan-apochromat objective on a Leica inverted SP5 or SP8 microscope with a laser scanning confocal system. Analysis was performed with ImageJ (http://imagej.nih.gov/ij/).

For co-localization experiments in HeLa cells, to remove soluble cellular proteins, cells were extracted with 0.03% saponin in cytosolic buffer (25 mM Hepes-KOH, pH 7.4, 25 mM KCl, 2.5 mM magnesium acetate, 5 mM ethylene glycol-bis(β-aminoethyl ether)-N,N,N′,N′-tetraacetic acid (EGTA), 150 mM K-glutamate) for 1 min prior to fixation. For co-localization analysis, regions of interest were drawn around individual cells. The Manders' coefficient was obtained using JACoP plugin and processed for statistical analysis with Prism (GraphPad software). Statistical significance was determined by non-parametric two-tailed *t* tests. Sample size was chosen arbitrary with no inclusion and exclusion criteria.

For IF analysis of Caco-2 cysts, cysts grown in matrigel in ibidi chambers were fixed in paraformaldehyde (PFA) 4% for 25 min, washed three times in PBS–glycine 0.75%, once in PBS, and permeabilized with 0.5% Triton X-100 for 20 min. Blocking was performed with 10% goat serum in IF buffer (0.1% BSA, 0.05% Tween-20, 0.2% Triton X-100, 0.05% NaN₃ in PBS). Samples were incubated overnight in appropriate primary antibodies dilutions in IF buffer at RT, then washed three times in IF buffer and incubated at RT for 1 h with secondary antibody dilutions 1:100 in IF buffer and 15 min with 4′,6-diamidino-2-phenylindole. Finally, samples were washed in PBS and post-fixed in PFA 4% for 10 min. Images were captured using a Leica inverted SP2 microscope with a laser scanning confocal system.

**Isoform detection by PCR.** Expression of myosin VI isoforms in cell lines or tissues was assessed by PCR. Messenger RNA from cultured cells or cysts was isolated with Maxwell RSC instrument (Promega) according to the manufacturer's protocols. RNA from mouse kidney and intestine tissues was obtained from C57BL/6 WT mouse and from commercially available human RNA (Takara Clontech Cat.No.: 636539 and 636529 for intestine and kidney, respectively). Retro-transcription was performed with the High Capacity cDNA Reverse Transcription Kit (Applied Biosystem). cDNA obtained was used in PCR reactions with primers flanking the spliced region as described in ref. [18].

**Protein expression and purification.** GST fusion proteins were expressed in *Escherichia coli* Rosetta (DE3) cells (Novagen) at 30 °C for 4 h after induction with 0.8 mM IPTG at an $OD_{600}$ of 0.6. Cell pellets were resuspended in lysis buffer

(50 mM Na-HEPES pH 7.5, 150 mM NaCl, 1 mM ethylenediaminetetraacetic acid (EDTA), 5% Glycerol, 0.1% NP40, Protease Inhibitor Cocktail set III, Calbiochem). Sonicated lysates were cleared by centrifugation at $48,000 \times g$ for 30 min. Supernatants were incubated with 1 ml of glutathione-sepharose beads (GE Healthcare) per liter of bacterial culture. After 4 h at 4 °C, the beads were extensively washed with PBS/0.1% Triton and equilibrated in storage buffer (50 mM Tris pH 7.4, 100 mM NaCl, 1 mM EDTA, 1 mM DTT, 10% glycerol). When needed, proteins were incubated overnight at 4 °C with PreScission Protease to obtain samples cleavaged from GST-tag. Cleaved proteins were further purified through Ion-Exchange chromatography on an ÄKTA Purifier FPLC system (GE Healthcare) and dialyzed overnight against storage buffer prior to snap-freezing in liquid nitrogen.

**Biochemical experiments.** For pull-down experiments, 1.5 µM of GST-fusion proteins immobilized onto GSH beads were incubated for 1 h at 4 °C in JS buffer (50 mM Hepes pH 7.5, 50 mM NaCl, 1.5 mM MgCl₂, 5 mM EGTA, 5% glycerol, and 1% Triton X-100) with either 1 mg lysate produced in JC buffer or the indicated concentration of cleaved and purified proteins. After extensive washes with JS buffer, beads were re-suspended in Laemmli-buffer and proteins analyzed through SDS-PAGE. Detection was performed either by staining the gels with Coomassie or by immunoblotting using specific antibodies. Ponceau-stained membrane was used to show loading of GST-fusion proteins.

For competition experiments, 1.5 µM of GST-CLCa immobilized onto GSH beads were incubated with 2.5 µM of His-Hip1R[346–655] for 1 hour at 4 °C in JS buffer, then increasing concentration (from 1.2 to 20 µM) of myosin VI[998–1131] were added and incubated for additional 30 min at 4 °C. Washes and detection as noted above.

For co-IP experiments, Caco-2 RFP-rat CLCa WT and I54D depleted of the endogenous CLCs as previously described, were lysed in JS buffer and IP was performed incubating 1 mg of lysate with RFP-Trap (Chromotek) for 120 min at 4 °C. After extensive washes with JS buffer, beads were re-suspended in Laemmli-buffer and proteins analyzed through sodium dodecyl sulfate-polyacrylamide gel electrophoresis (SDS-PAGE) and immunoblotting.

FP assays were carried out at 22 °C on a 384-well plate with Infinite 200 instrument (Tecan) using an excitation wavelength of 535 nm and an emission wavelength of 580 nm. Concentrations in the nanomolar range (20–50 nM) of 5-FAM-conjugated CLCa peptides (GenScript) were titrated with the indicated myosin VI-purified fragment starting from an initial concentration of ~150 µM. Polarization readings from three independent experiments were averaged and fitted as described in Eletr et al.[62].

For analytical size exclusion chromatography, 50 µM CLCa[46–61] was incubated at 1:1 molar ratio with 50 µM of myosin VI[998–1131] fragment for 15 min at 4 °C and subjected to size exclusion chromatography on a Superdex75 (5/150) column using the ÄKTA microsystem (GE Healthcare). Fractions containing complex or single proteins were analyzed by SDS-PAGE and Coomassie staining.

ITC measurements were performed in a buffer containing 20 mM Tris-HCl pH 8.0, 150 mM NaCl, and 2.5% glycerol, at 25 °C on a MicroCal PEAQ-ITC (Malvern Panalytical, Malvern, UK) instrument. All protein samples were dialyzed overnight in this buffer prior to titration. In all, 380 µM CLCa WT or I54D mutant were titrated into 20 µM of either Hip1R[346–655] or myosin VI[1050–1131] provided in the reaction cell. For each experiment, we used 19 injections of 2 µl with 150 s spacing and 750 rpm stirring speed. Measurements were performed in duplicates or triplicates. Raw data were analyzed with the integrated Malvern analysis software, and heat production was fitted to a one-set-of-sites binding model.

**CCV preparation from porcine brain.** Fresh porcine brain (First Link Ltd, Birmingham) was thawed in 100 ml PBS per 50 g tissue at 37 °C. Once thawed, PBS was removed and substituted with buffer A (100 mM 2-(N-morpholino)ethanesulfonic acid, 1 mM EGTA, 0.5 mM MgCl₂) containing 2 mM phenylmethylsulfonyl fluoride (PMSF). Homogenization was performed at 4 °C in a mixer, at pulse mode to avoid warming of the mixture. The homogenate was then centrifuged at $8000 \times g$ for 30 min at 4 °C in a JA-17 rotor (Beckman Coulter) and supernatant filtered through two layers of gauze bandage to further remove particles. The final filtrate was pooled and centrifuged at $185,000 \times g$ for 60 min at 4 °C in a Ti-45 rotor (Beckman Coulter) to pellet CCVs. To further purify CCVs from impurities, the pellet was resuspended in a final volume of 12–15 ml buffer A containing 2 mM PMSF and 0.02% NaN₃ and homogenized on ice for 5 min using a SS30 dounce stirrer (Stuart) at 125 rpm. An equivalent volume of a 12.5% Ficoll and 12.5% sucrose solution was added to the CCV homogenate and mixed gently. The mixture was then centrifuged at $30,000 \times g$ for 45 min at 4 °C using a JA-17 rotor (Beckman Coulter). CCV-containing supernatant was diluted fivefold with buffer A + 2 mM PMSF and CCVs sedimented at $185,000 \times g$ for 60 min at 4 °C in a Ti-45 rotor (Beckman Coulter). The supernatant was discarded, vesicles resuspended in 4 ml buffer A + 2 mM PMSF and 0.02% NaN₃, aliquoted and snap-frozen in liquid nitrogen, and stored at −80 °C until further use[63].

**Purification of triskelia from CCVs.** Tris-extraction was performed according to ref. [63] with slight modifications. Purified CCV were quickly thawed at 37 °C and

kept on ice. A solution of Tris pH 7.3, EDTA, protease inhibitor cocktail, and β-mercaptoethanol was added to the CCV suspension to obtain the final concentration of the Tris-extraction buffer (0.5 M Tris pH 7.3, 2 mM EDTA, 1 mM β-mercaptoethanol and 1 × protease inhibitors). The mix was left on ice for 10 min before ultracentrifugation in a TLA 100.3 rotor (Beckman Coulter) for 30 min at $220,000 \times g$ at 4 °C. Supernatant containing clathrin triskelia and adaptor proteins was further purified via size exclusion chromatography in buffer B (0.5 M Tris pH 7.4, 2 mM EDTA) using a Superose 6 increase 10/300 GL column (GE Healthcare) connected to an ÄKTA Purifier FPLC system (GE Healthcare) at a flow rate of 0.5 ml/min at 8 °C. Elution fractions of 0.5 ml volumes were collected and analyzed via SDS-PAGE. Clean clathrin fractions were pooled, and an equivalent volume of saturated NH₄SO₄ solution was carefully added. The mix was incubated on ice for 30 min to allow proteins precipitation and centrifuged for 30 min at $1600 \times g$ using an Eppendorf 5810 R centrifuge (Eppendorf) at 4 °C. The pellet was dissolved in 70–150 µl extraction buffer (depending on pellet size), dialyzed in buffer B for 1 h, and overnight at 4 °C in buffer C (50 mM Tris pH 8.0, 50 mM NaCl, 2 mM EDTA, 1 mM β-mercaptoethanol, 1 mM PMSF). Protein concentration (3–5 mg/ml) was determined by ultraviolet (UV) absorbance at 280 nm, using an extinction coefficient of 1.0 (mg/ml)/cm. Clathrin triskelia were either directly subjected to CLC dissociation (see below) or dialyzed against buffer A containing 2 mM CaCl₂ overnight at 4 °C to allow cage assembly and long-term storage as native cages.

**CLC dissociation and cage re-assembly.** CLC dissociation was performed as previously described (Winkler and Stanley 1983). Clathrin triskelia solution were diluted in stripping buffer D (50 mM Tris pH 8.0, 50 mM NaCl, 1.3 M NaSCN, 2 mM EDTA, 1 mM β-mercaptoethanol, 1 mM PMSF) and subjected to size exclusion chromatography on a Superose 6 increase 10/300 GL column (GE Healthcare). Elution fractions were collected and analyzed via SDS-PAGE. Clean clathrin fractions were pooled, dialyzed twice against buffer C to remove NaSCN and concentrated by spin filtration using 10 K Amicon-Ultra centrifugal filters (Merck Millipore). Protein concentrations were determined by UV absorbance at 280 nm using an extinction coefficient of 1.0 (mg/ml)/cm. To assemble CHC-only cages, CHC triskelia were dialyzed overnight in buffer A containing 2 mM CaCl₂. To generate CHC-CLC specific cages, CHC-only cages were reconstituted with either CLCa or CLCb purified proteins in slight molar excess (CHC:CLC 1:1.2). Mix were incubated on ice for 1 h, centrifuged for 30 min at 4 °C in a TLA55 rotor at $150,000 \times g$ and washed once with buffer A to eliminate the unbound CLCs.

**Co-sedimentation of clathrin cages with myosin.** Native and CLCa or CLCB-only cages were diluted to 1.5 µM in buffer A + 0.1% Triton X-100 and 0.1 mg/ml BSA and mixed with 1.5 µM of purified myosin VI[998–1131] fragment on ice for 45 min. Mix were then centrifuged at 4 °C for 30 min in a TLS-55 rotor at $108,000 \times g$, pellets were washed once with same volume of buffer A + 0.1% Triton X-100 and 0.1 mg/ml BSA and centrifuged. Finally, pellets were dissolved in Laemmli buffer and analyzed through SDS-PAGE.

**Negative staining of clathrin cages.** Cages were adsorbed to freshly glow-discharged, carbon-coated formvar films on copper grids by placing grids on 10 µl droplets of samples at 0.2 mg/ml on Parafilm for 90 s. Excessive liquid was removed using filter paper. Grids were washed twice by sequentially transferring grids onto 15 µl droplets of buffer A containing 2 mM CaCl₂. Samples were then stained with 2% uranyl acetate in water for 1 min and air-dried before subjected to EM analysis. Specimens were observed using a Tecnai G2 (FEI) electron microscope at an acceleration voltage of 120 kV. Images were obtained using a SIS Morada digital camera and TIA software (FEI).

**EM of Caco-2 cysts.** Cysts grown in matrigel were fixed with 2.5% glutaraldehyde in 0.1 M sodium cacodylate buffer pH 7.4 for 1 h at RT, washed three times with cacodylate buffer, and fixed in 1% osmium tetroxide and 1.5% potassium ferrocyanide in 0.1 M cacodylate for 1 h on ice. After several washes in distilled water, samples were en bloc stained with 0.5% uranyl acetate in water overnight at 4 °C. Finally, samples were dehydrated in a graded ethanol series (30%, 50%, 70%, 80%, 90%, 96%, 20 min each and 3 washes with absolute ethanol, 10 min each). Next, the samples were infiltrated in a 1:1 ethanol/Epon 812 solution for 2 h, in 100% Epoxy resin twice for 1 h each, and left overnight in resin before being embedded in resin and polymerized at 60 °C for 48 h. At the end of polymerization, the resin blocks were separated from the multiwell chamber and part of the block containing few cysts was mounted on a Leica Ultracut UCT ultramicrotome. Ultrathin (70–90 nm) sections were then collected on formvar carbon-coated slot grids and stained with uranyl acetate and Sato's lead citrate before imaged with a Talos 120 C (FEI) electron microscope; images were acquired with a 4k × 4 K Ceta CMOS camera.

For morphometrical analysis of the distribution of clathrin-coated structures, between 94 and 125 cellular profiles of well-polarized cells for each biological sample were imaged. Images of the CCPs present on the apical surface and connected to the membrane were acquired at the nominal magnification of ×36,000 and classified as shallow, omega/constricted, or elongated according to their morphology. The distribution of clathrin-coated structures has been expressed as

percentage among the different classes (Fig. 5e) or as the number per microns of apical membrane (Supplementary Fig. 10g). All statistical analyses were performed using GraphPad Prism. One-way analysis of variance was used to calculate statistical significance among different samples.

**Protein expression and purification for NMR spectroscopy.** GST-tagged myosin VI and CLCa were expressed separately in *E. coli* BL21 (DE3) cells (Thermo Fischer Scientific) at 17 °C overnight upon reaching an $OD_{600}$ value of 0.5–0.6 by induction with 0.4 mM IPTG. Cell pellets were resuspended in Buffer A (50 mM HEPES, pH 7.5, 300 mM NaCl, 1 mM EDTA, 5% glycerol, 2 mM DTT, and Roche Complete Mini protease inhibitor cocktail), lysed by sonication, and the lysates were cleared via centrifugation. Supernatants were incubated with glutathione sepharose beads (GE Healthcare) for 3 h at 4 °C with a fresh protease inhibitor cocktail tablet added. Myosin VI-bound beads were washed twice with Buffer A, twice with Buffer B (PBS plus 500 mM NaCl, 0.1% Triton X-100, and 2 mM DTT, pH 7.4), twice with Buffer C (PBS plus 2 mM DTT, pH 7.4), and once with Buffer D (50 mM Tris-Cl, 100 mM NaCl, 1 mM EDTA, 10% glycerol, and 2 mM DTT, pH 7.0). The washed beads were resuspended in Buffer A and incubated with PreScission Protease (GE Healthcare) overnight at 4 °C. Myosin VI was eluted with Buffer A, concentrated to 2 ml, purified via size exclusion chromatography on an FPLC system (Superdex75, GE Healthcare) with FPLC Buffer (20 mM $NaPO_4$, 50 mM NaCl, 2 mM DTT, pH 6.5), and concentrated to 1.5 ml. CLCa-bound glutathione sepharose beads were washed five times with Buffer A, at which point purified myosin VI (from the above purification protocol) was added to the beads. The CLCa–myosin VI mixture was incubated with PreScission Protease overnight and eluted with Buffer A. Elutions were concentrated to 2 ml, purified via size exclusion chromatography on an FPLC system with FPLC Buffer, and concentrated to ~300 μl.

**NMR spectroscopy and backbone assignment.** NMR data were collected at 10 °C on Bruker 700 and 850 MHz spectrometers. All experiments were conducted in 20 mM $NaPO_4$ pH 6.5, 50 mM NaCl, 2 mM DTT, 0.1% $NaN_3$, 1 mM Pefabloc, and 5% $D_2O$. In all, 0.78 mM $^{15}N$-labeled myosin VI and synthesized 5-FAM N-terminally tagged CLCa (GenScript) were used for the HSQC experiments acquired for CSP analysis. For structure determination, two separate samples were produced, namely, 0.4 mM $^{13}C$, $^{15}N$-labeled CLCa with equimolar unlabeled myosin VI, and 0.36 mM $^{13}C$, $^{15}N$-labeled myosin VI with equimolar unlabeled CLCa. For each sample, the following NMR experiments were performed: $^{15}N$-HSQC, $^{13}C$-HSQC, HNCO, HNCACB, $^{15}N$-dispersed NOESY (120 ms mixing time), $^{13}C$-dispersed NOESY (100 ms mixing time), and $^{13}C$-dispersed-half-filtered-NOESY (100 ms mixing time). The NMR spectra were processed with NMRpipe[64] and analyzed with XEASY[65]. Backbone assignments of myosin VI and CLCa in complex were done manually based on chemical shifts obtained previously on free myosin VI (BMRB 25544) and the HNCACB and HNCO spectra; the HNCACB spectrum showed both the intra-residue and proceeding residue Cα signals for most residues (Supplementary Fig. 4). Residues that were ambiguous were verified with HN-HN and α-HN NOEs in the $^{15}$NOESY (Supplementary Fig. 5). Sidechain assignments were obtained by manual analysis of HNCACB, $^{13}C$-dispersed NOESY, and $^{15}N$-dispersed NOESY experiments.

**Structure determination and refinement.** NOE assignments were done manually using $^{13}C$-dispersed NOESY, $^{15}N$-dispersed NOESY, and $^{13}C$-dispersed half-filtered NOESY experiments. Chemical shifts for backbone carbons and amide nitrogen, amide hydrogen, Hα, and Cβ from myosin VI and CLCa were used with TALOS[66] to obtain backbone ϕ and ψ torsion angle restraints. Structure calculations were performed with Xplor-NIH (version 2.47)[67] by using standard scripts. Twenty linear starting structures were subjected to 35,000 simulated annealing steps followed by 5000 cooling steps, both of 0.005 ps. The lowest energy structure from this set was used as the starting structure for refinement closely following the protocol described in ref. [68] but with a temperature step of 1 K used during simulated annealing. Constraints used during the structure calculations include intramolecular and intermolecular NOE restraints, dihedral angle restraints, and backbone hydrogen bonds in helical regions. A summary of the data used for the structure calculations is provided in Table 1. Hydrogen bond distances between the acceptor oxygen and donor hydrogen and nitrogen atoms were set to 1.81–2.09 and 2.71-3.13 Å, respectively. Structures and NOE assignments were checked manually to identify errors, corrections were made if necessary, and structures recalculated; this process was done iteratively. Of the 300 final refined structures calculated, the 20 lowest energy structures without distance violations >0.5 Å and dihedral angle violations >5° were selected to represent the structure of myosin VI in complex with CLCa. Pymol (Schrödinger) was used for visualization and figure generation; MOLMOL[69] was used for r.m.s.d. calculation and figure generation. Among the 20 lowest energy structures, no amino acid was disfavored by Ramachandran analyses; 1, 6, and 94% were in the generously allowed, allowed, and favored regions, respectively.

**Reporting summary.** Further information on research design is available in the Nature Research Reporting Summary linked to this article.

## Data availability

Data supporting the findings of this manuscript are available in the Source Data file. All uncropped immunoblots associated with Figs. 1b, 1c, 1d, 1e, 2b,4f, 4d, 5a, 6b, 6c and Supplementary Figs. 1a, 1c, 2d, 9a, 9b, 10a,10c, 11c, 11d, 11e, 12a, 12b are provided as Source Data file. A reporting summary for this Article is available as a Supplementary Information file. Chemical shift assignments for the complex of myosin VI$^{1050–1131}$ and CLCa$^{46–61}$ have been deposited in the Biological Magnetic Resonance Bank under the ID code 30500, and the atomic coordinates for the complex have been deposited in the Protein Data Bank under the ID code 6E5N.

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

## Acknowledgements

We thank Tom Kirchhausen, Stephen Royle, Klemens Rottner, and Elizabeth Smythe for DNA constructs. We thank ALEMBIC facility at San Raffaele Scientific Institute, Milan, Italy for support in electron microscopic analysis and the Biochemistry and Structural Biology Unit at the European Institute of Oncology for support with isothermal titration calorimetric analysis. This work was supported by the Associazione Italiana per la Ricerca sul Cancro (AIRC IG19875 to S.P.); the Italian Ministry of Education, Universities and Research (PRIN 20108MXN2J to S.P.); the Intramural Research Program through the CCR, NCI, NIH (1ZIABC011627 to K.J.W.); and Wellcome Trust Investigator Award (107858/Z/15/Z to F.M.B.). C.A.N.'s work is supported by the Marie Skłodowska-Curie Actions (MSCA-IF-2016 #752553). M.B. was and R.S.d.P. is a PhD student within the European School of Molecular Medicine (SEMM). M.B was supported by a fellowship from Fondazione Umberto Veronesi (FUV) and by an EMBO short-term travel fellowship.

## Author contributions

M.B. performed most of the biochemical experiments and contributed to result interpretation; G.R.B. performed the NMR experiments and solved the structures; C.A.N generated the genetically reconstituted Caco-2 cells and cysts and contributed to experimental design; E.M. performed the Hip1R experiments; R.S.d.P. prepared the samples for EM analysis; A.R. conducted the EM experiments; L.R. supervised the mice experiments; J.W. performed the ITC experiments. F.M.B. generated the CLC KO mice and contributed to experimental design and paper writing; K.J.W. supervised the NMR analysis and contributed to NMR interpretation and paper writing; S.P. coordinated the team, designed the experiments, and wrote the paper.

## Competing interests

The authors declare no competing interests.

**Additional information**

