## [Peer Review File · Nature Communications]

Reviewers' Comments:

Reviewer #1:

Remarks to the Author:

The manuscript demonstrates a rather tight, specific interaction between a peptide segment of CLCa and the alpha2-isoform of myosin-VI, involving both the alpha2 helix and the C-terminal cargo-binding domain of myosin-VI. The study provides evidence for the Ca²⁺ dependence of the interaction by co-immuno precipitation, attributed to a locked conformation of the CBD induced by apo-calmodulin binding, which is abolished upon Ca-ligation of calmodulin. The study presents interesting new information on the role of CLCa in recruiting myosin-VI in formation/fusion of clathrin-coated pits, an area of high interest in molecular biophysics.

The detailed interaction surface between CLCa and myosin VI appears to nicely explain the specificity of CLCa over CLCb binding, as well as the binding competition with Hip1R. Interestingly, the 46-61 peptide segment of CLCa appears disordered in the absence of myosin VI, and provides yet another example of the importance of such intrinsically disordered regions in regulating pivotal cellular processes.

The structural conclusions rely extensively on multinuclear multi-dimensional triple resonance NMR spectroscopy, using well established and robust isotope-filtering NOESY techniques to identify a quite large set of intermolecular NOE contacts between the 15-residue CLCa peptide and the C-terminal 3-helix domain for which they previously reported the NMR structure (PDB 2N12). Chemical shift perturbation agrees with the observed NOE contacts, and the quality of the structure appears to be as good as can be obtained by conventional NOE-based methods. Experimental data are extensively documented in the SI and universally appear of high quality. A very minor technical issue one might quibble about is whether the reported change in orientation of alpha4 is real, or whether the CSP simply reflect the intermolecular interactions involving W1124 and the experimentally observed difference in alpha4 orientation simply reflects experimental uncertainty. It is widely appreciated that the accuracy of NMR structures is lower than suggested by the width of the "NMR bundle of structures", and without some more global restraints (such as RDCs) it is quite difficult, if not impossible, to evaluate whether small apparent rearrangements of a helix such as seen for alpha4 are real or simply reflect the experimental uncertainty. This minor point has no bearing on any of the conclusions, and I therefore do not suggest the recording of additional RDC data, simply a slightly more subtle wording of this minor detail.

Reviewer #2:

Remarks to the Author:

This work investigates the interaction between two isotopes of the clathrin light chain (CLCa and b) with myosin VI, using a range of biochemical, cell biological and structural approaches. The work begins with interaction assays for the long and short isoforms of myosin VI with CLCa and CLCb demonstrating that CLCa interacts with the long form of myosin VI (Fig. 1). I would prefer it if the diagram of myosin VI in Fig. 1A additionally showed a full length diagram of myosin VI, as the first thing I needed to do was remind myself where residue 998 was in respect to the motor, IQ, 3 helix bundle and single alpha helical domain of myosin VI, particularly as I was briefly confused as to whether the alpha helices being described were part or not part of the single alpha helical domain. Further work goes on to describe the interaction between CLCa and Myosin VI, demonstrating that the alpha2 helix is needed. The author then investigate the structure (Fig.3) by NMR, and interaction between myosin VI and CLCa. Fig. 3F refers to labels in grey and black, which were not so easy to spot - why not use red and black instead?

The authors, having nicely defined how CLCa and myosin VI interact in detail, then switch back to cell biology, testing out mutations to firm up their conclusions, focusing in on the CLCa mutant 154D, which abolishes the interaction between CLCa and myosin VI. They go on to look at the

localisation of wild type and mutant CLCa in Caco2 cysts (Fig. 5d) - which is somewhat difficult to see the staining in - I suggest individual colours are shown in greyscale for better contrast. A scale bar is also needed. In the final panel of Fig. 5, the authors have characterised the formation of clathrin coated pits by electron microscopy, however although this states that two independent experiments were done, there is no information on how many pits were counted, or how many cells, and this cannot be guessed at as the numbers are presented in %. I would have expected perhaps for the three different phenotypes of pit - to be counted per cell, across multiple cells, and two experiments with some statistics here to show key differences.

Finally, the authors move on to investigate Hip1R (incidentally, this needs spelling out in full in the abstract), and how it might compete with CLCa for binding to myosin VI.

Overall, the paper provides new insight into the interaction of CLCs with myosin VI, which will be of interest to a broad audience.

The authors then go onto speculate about how myosin VI might be recruited (and when) to the CCP (Fig. 6d). I think this is the only part of the paper that I have an issue with as quite a bit of this is speculative and not wholly based on the experiments presented in the paper. To genuinely test that hypothesis, one would have to see images of CCPs (e.g. super-resolution STORM or STED) that justify the images in the diagram, as to the localisation of the different proteins. Moreover, the jury is still out on whether Dab2 does actually dimerise myosin VI or not. The crystallography paper cited uses a construct with both Myosin VI tail domain and a Dab2 peptide on the same polypeptide and in the crystal, the Dab2 peptide binds to the tail of a second molecule of myosin VI, not the same molecule on the same polypeptide, as you might expect. In that paper, cross-linking had to be used to obtain an interaction between these two proteins, in solution. Work from the Buss lab (Spudich et al., Nature Cell Biology, 2007) also hedge their bets on whether there is dimerisation or not, and recent reviews from the same lab are equally non committal. I would argue that there is no need for myosin VI to dimerise in this context, as it could equally work in teams, rather than as a dimer (e.g. see work from Sivaramakrishnan and Spudich, J. Cell Biol., 2009). A revision of the model and the discussion needs to reflect these criticisms and our ongoing uncertainty about myosin VI dimerisation.

Other minor comments:

Scale bars for supplementary Fig. 1 b (for 'zoomed' in views) are missing Ditto Fig. 2C. Fig. 10b and e - missing scale bar

Reviewer #3:

Remarks to the Author:

The manuscript by Biancospino et al. reports the characterisation of the interaction between the long splicing variant of Myosin-VI (Myosin-VI) and clathrin-light-chain a (CLCa). First the authors define the molecular specificity of the association of Myosin-VI-long with CLCa in clathrin triskelions and cages by performing binding assays and co-localization experiments in HeLa cells. Then they map the minimal binding interfaces required to reconstitute the interaction, spanning residues 998-1131 of Myosin-VI-long and residues 46-61 of CLCa. To understand the topology of the interaction, they determine the solution structure of the complex, that shows an isoform-specific three helical bundle of Myosin-VI harbouring a short helical stretch of CLCa. Structure analysis identifies key residues on both sides of the interface required for the binding, including CLCa-I54. The mutation CLCa-I54D is introduced in Caco-2 cells and used to test the phenotypic effect of impairing the Myosin-VI/CLCa interaction in polarised epithelia. Finally, by sequence analysis, the authors identify a Myosin-VI-like motif in Hip1R implicated in CLCa binding in vitro.

The manuscript characterises the Myosin-VI/Cathrin interaction, providing molecular information on the binding interface and on the functional implications for clathrin activities in polarised epithelial cells. The findings are novel and well presented, and uncover an interesting isoform-specific function of Myosin-VI that will be interesting for scientists in the field. However, a few issues remain open on the biological relevance of the interaction. I am in favour of publication in Nature Communications provided that these issues are addressed, as outlined below.

Major issues

1. To evaluate the relevance of the Myosin-VI-long/CLCa interaction the authors generate Caco-2 cell lines stably expressing CLCa-WT and the I54D mutant. Analysing three-dimensional cysts grown with these cell lines ablated of endogenous CLCa, they observe morphologically defective clathrin coated pits but no polarity defects. Here some controls are needed.

- In Fig 5c the authors show an immuno-blot of RFP-CLCa in the cell lines used for the rescue experiments. It would be informative to show the relative levels of endogenous CLCa and ectopically expressed rescue constructs in polarised Caco-2 cells in an IB with the same exposure. If all exposures are already equal, the authors should comment on the fact that the rescue constructs are expressed at lower levels than the endogenous protein (and possibly repeat experiments with higher rescue levels), and that CHC levels seems reduced upon CLC depletion.

- In Fig 5d, representative images of cysts grown with ctrl and Caco-2 rescue cell lines are shown. The authors conclude that no polarity nor morphogenetic defects are visible. However E-cadh seems more apical in the rescue lines, both the one expressing CLCa-WT and the one expressing the I54D mutant. In addition, cysts with these cell lines show multilayering and cells inside the lumen. The authors should comment on this evidence, and stain the cysts with additional polarity markers (for instance aPKC, actin, DLG...) to ensure that apico-basal polarity is properly rescued and preserved in the I54D mutant cells. It would also be interesting to monitor the localisation of Myosin-VI in the cysts with CLCa rescue constructs.

- In Fig 5e, EM analysis of Caco-2 cysts show that the morphology of CCPs is impaired in the CLCa-I54D mutant. In these experiments the lumen and the apical side of Caco-2 cells is visualised in cysts treated with cholera toxin, which induces lumen expansion and might affect membrane tension. It would be useful to repeat the experiment in cysts without CTX treatment.

- Also, if possible, it would be important to know if the defective morphology observed in the CCPs of cysts with CLCa-I54D results in defective endocytosis, or can be tolerated / bypassed.

2. In Fig 6, the authors identify an amino acidic stretch (IENDEA) in Hip1R partly overlapping with the CLCa-binding region of Myosin-VI. This stretch binds to CLCa in vitro and is out-competed by Myosin-VI-1050-1131. Notably, the IENDE motif is also present in CLCb and mediate binding of Hip1R to CLCb, possibly with higher affinity based on Supplementary Fig. 10h. The relevance of the competitive interaction between Hip1R and Myosin-VI in binding to CLCa for clathrin-mediated endocytosis is not worked out further in cells. Thus, it remains speculative. The authors should analyse the interplay between Myosin-VI and Hip1R in CLC-a/b binding in cells, or move this information to the supplementary material.

Minor points

- Experimental errors bars should be added to FP graph shown in Fig. 2c-2d and Supplementary Fig 9b.

- To make the structure of the Myosin-VI/CLCa-46-61 easier to understand, the authors should consider illustrating the structure with same colors of the cartoons in Fig 1a-2a, and with orientations related to one another, including the enlarged views. Also, it might be beneficial to organise the panels in order to have the validation of the interface shown in Fig 5a-5b included in Fig. 4 together with the close-up views.

- In supplementary Fig 1a no asterisk is visible. The figure or the caption should be modified to account for this discrepancy.

- Molecular Weight Marker should be added in the blot/SDS-PAGES presented in Supplementary Fig 2d.

- In Fig 1 legend and supplementary Fig 9a, Myosin-VI-short seems to have the same numbering of Myosin-VI-long (ending at residue 1131). This is somehow confusing, it would be better to modify it.

- Supplementary Fig 10f shows the KO efficiency of siRNA targeting CLCa/b. Based on the plots, in cysts the KO-efficiency after 7 days is about 60% compared to CTRL cysts, which is not very high. The authors should comment on these data, and on the choice of using transient silencing for the cysts experiments.

Reviewer #1:

The manuscript demonstrates a rather tight, specific interaction between a peptide segment of CLCa and the alpha2-isoform of myosin-VI, involving both the alpha2 helix and the C-terminal cargo-binding domain of myosin-VI. The study provides evidence for the Ca²⁺ dependence of the interaction by co-immuno precipitation, attributed to a locked conformation of the CBD induced by apo-calmodulin binding, which is abolished upon Ca-ligation of calmodulin. The study presents interesting new information on the role of CLCa in recruiting myosin-VI in formation/fusion of clathrin-coated pits, an area of high interest in molecular biophysics.

The detailed interaction surface between CLCa and myosin VI appears to nicely explain the specificity of CLCa over CLCb binding, as well as the binding competition with Hip1R. Interestingly, the 46-61 peptide segment of CLCa appears disordered in the absence of myosin VI, and provides yet another example of the importance of such intrinsically disordered regions in regulating pivotal cellular processes.

R. We thank the Reviewer for his/her nice words.

The structural conclusions rely extensively on multinuclear multi-dimensional triple resonance NMR spectroscopy, using well established and robust isotope-filtering NOESY techniques to identify a quite large set of intermolecular NOE contacts between the 15-residue CLCa peptide and the C-terminal 3-helix domain for which they previously reported the NMR structure (PDB 2N12). Chemical shift perturbation agrees with the observed NOE contacts, and the quality of the structure appears to be as good as can be obtained by conventional NOE-based methods. Experimental data are extensively documented in the SI and universally appear of high quality. A very minor technical issue one might quibble about is whether the reported change in orientation of alpha4 is real, or whether the CSP simply reflect the intermolecular interactions involving W1124 and the experimentally observed difference in alpha4 orientation simply reflects experimental uncertainty. It is widely appreciated that the accuracy of NMR structures is lower than suggested by the width of the "NMR bundle of structures", and without some more global restraints (such as RDCs) it is quite difficult, if not impossible, to evaluate whether small apparent rearrangements of a helix such as seen for alpha4 are real or simply reflect the experimental uncertainty. This minor point has no bearing on any of the conclusions, and I therefore do not suggest the recording of additional RDC data, simply a slightly more subtle wording of this minor detail.

R. Agree. We thank the Reviewer for pointing out this subtle but important point. We looked back at the data for the free myosin VI structure and agree that there is some ambiguity to this point. We have therefore deleted this discussion and agree that the myosin VI structure is largely unchanged upon binding to CLCa. We have therefore revised Figure 4 to remove the overlay while still showing the hydrogen bond between W1124 and E50.

Reviewer #2:

This work investigates the interaction between two isotopes of the clathrin light

chain (CLCa and b) with myosin VI, using a range of biochemical, cell biological and structural approaches. The work begins with interaction assays for the long and short isoforms of myosin VI with CLCa and CLCb demonstrating that CLCa interacts with the long form of myosin VI (Fig. 1). I would prefer it if the diagram of myosin VI in Fig. 1A additionally showed a full length diagram of myosin VI, as the first thing I needed to do was remind myself where residue 998 was in respect to the motor, IQ, 3 helix bundle and single alpha helical domain of myosin VI, particularly as I was briefly confused as to whether the alpha helices being described were part or not part of the single alpha helical domain.

R. We thank the reviewer for drawing our attention to this lack of clarity. We have now added the full-length diagram of myosin VI to better explain the region of CLCa binding and its relative position.

Further work goes on to describe the interaction between CLCa and Myosin VI, demonstrating that the alpha2 helix is needed. The author then investigate the structure (Fig.3) by NMR, and interaction between myosin VI and CLCa. Fig. 3F refers to labels in grey and black, which were not so easy to spot - why not use red and black instead?

R. We thank the Reviewer for pointing out that the labels were difficult to see; we have now made the suggested modification and agree that the figure is improved.

The authors, having nicely defined how CLCa and myosin VI interact in detail, then switch back to cell biology, testing out mutations to firm up their conclusions, focusing in on the CLCa mutant 154D, which abolishes the interaction between CLCa and myosin VI. They go on to look at the localisation of wild type and mutant CLCa in Caco2 cysts (Fig. 5d) - which is somewhat difficult to see the staining in - I suggest individual colours are shown in greyscale for better contrast. A scale bar is also needed.

R. We thank the reviewer for bringing this lack of clarity to our attention. We have now added individual colors in greyscale for better viewing contrast and added the scale bar, as requested.

In the final panel of Fig. 5, the authors have characterised the formation of clathrin coated pits by electron microscopy, however although this states that two independent experiments were done, there is no information on how many pits were counted, or how many cells, and this cannot be guessed at as the numbers are presented in %. I would have expected perhaps for the three different phenotypes of pit - to be counted per cell, across multiple cells, and two experiments with some statistics here to show key differences.

R. We thank this reviewer for making us aware of this ambiguity in our description of our data. The number of cellular profiles of well polarized cysts were actually mentioned in the legend of the Supplementary Figure 10g (between 70-100). In any case, we performed a third experiment that confirmed our previous results. We have now added to the new Figure 5c the number of pits counted and - in the legend- the number of cellular profiles analyzed. The new data are also added to the panel d of the new Supplementary Figure 11 together with the required

statistics.

Finally, the authors move on to investigate Hip1R (incidentally, this needs spelling out in full in the abstract), and how it might compete with CLCa for binding to myosin VI. Overall, the paper provides new insight into the interaction of CLCs with myosin VI, which will be of interest to a broad audience.

R. We thank the Reviewer for his/her enthusiasm towards our findings and also for their suggestions, which we agree have improved our manuscript. In addition, as suggested, we have now spelled out Hip1R in the abstract.

The authors then go onto speculate about how myosin VI might be recruited (and when) to the CCP (Fig. 6d). I think this is the only part of the paper that I have an issue with as quite a bit of this is speculative and not wholly based on the experiments presented in the paper. To genuinely test that hypothesis, one would have to see images of CCPs (e.g. super-resolution STORM or STED) that justify the images in the diagram, as to the localisation of the different proteins.

R. This part of the manuscript is in the Discussion section and was meant not as a final recap of what we discovered, but rather as a Discussion section often is, as a thoughtful integration of where the field is now and what future directions are needed.

We have now clarified this limitation of our model in our new version of the manuscript, directly stating that future experiments are needed to test the hypotheses. Moreover, we added as a reference, a paper from Merrifield and colleagues that analysed the time of recruitment of myosin VI at the clathrin-coated structures by total internal reflection fluorescence microscopy (TIRF) with a resolution of 2 s. In line with our model, myosin VI is classified as part of the scission module together with dynamin and regulators of actin polymerization (Taylor MJ. et al PLoS Biology 2011). We are actively working on the localization of the implicated proteins by super-resolution, but imaging of CCPs in 3D culture is extremely complex and will require a large amount of effort that is beyond the scope of this current manuscript.

To strengthen our data on Hip1R, we have now measured binding affinities by ITC experiments with CLCa full-length WT and I54D mutant proteins. The data show that Hip1R binds CLCa with a log lower affinity with respect to myosin VI and that the single point mutation in CLCa abrogates binding to myosin VI while leaving unaffected Hip1R binding (see new Fig. 6d). The difference in affinity corroborates our hypothesis that Hip1R could be displaced by myosin VI during CCP maturation, following calcium mobilization (new Supplementary Fig. 12).

Moreover, the jury is still out on whether Dab2 does actually dimerise myosin VI or not. The crystallography paper cited uses a construct with both Myosin VI tail domain and a Dab2 peptide on the same polypeptide and in the crystal, the Dab2 peptide binds to the tail of a second molecule of myosin VI, not the same molecule on the same polypeptide, as you might expect. In that paper, cross-linking had to be used to obtain an interaction between these two proteins, in solution. Work from the Buss lab (Spudich et al., Nature Cell Biology, 2007) also hedge their bets on whether there

is dimerisation or not, and recent reviews from the same lab are equally non committal. I would argue that there is no need for myosin VI to dimerise in this context, as it could equally work in teams, rather than as a dimer (e.g. see work from Sivaramakrishnan and Spudich, J. Cell Biol., 2009). A revision of the model and the discussion needs to reflect these criticisms and our ongoing uncertainty about myosin VI dimerisation.

R. We thank the reviewer for raising these points as indeed we are aware of the debated issue on the Dab2 ability to dimerize myosin VI. Recent literature seems to corroborate the idea that various interactors could promote dimerization or even oligomerization of myosin VI (Phichith D et al PNAS 2009, Shang G et. al Elife 2017 and references therein).

Accordingly, in her recent review (de Jonge JJ, FEBS letter 2019), Folma Buss described Dab2 and Optineurin as a dimer inducer and depicted them in Figure 1 as we did in our model. We have revised our discussion to integrate these points and thank the reviewer.

Other minor comments:

Scale bars for supplementary Fig. 1 b (for 'zoomed' in views) are missing Ditto Fig. 2C. Fig. 10b and e - missing scale bar

R. We thank the reviewer for bringing this to our attention. We have now revised the figure panels mentioned and added additional information to the legend.

Reviewer #3:

The manuscript by Biancospino et al. reports the characterisation of the interaction between the long splicing variant of Myosin-VI (Myosin-VI) and clathrin-light-chain a (CLCa). First the authors define the molecular specificity of the association of Myosin-VI-long with CLCa in clathrin triskelions and cages by performing binding assays and co-localization experiments in HeLa cells. Then they map the minimal binding interfaces required to reconstitute the interaction, spanning residues 998-1131 of Myosin-VI-long and residues 46-61 of CLCa. To understand the topology of the interaction, they determine the solution structure of the complex, that shows an isoform-specific three helical bundle of Myosin-VI harbouring a short helical stretch of CLCa. Structure analysis identifies key residues on both sides of the interface required for the binding, including CLCa-I54. The mutation CLCa-I54D is introduced in Caco-2 cells and used to test the phenotypic effect of impairing the Myosin-VI/CLCa interaction in polarised epithelia. Finally, by sequence analysis, the authors identify a Myosin-VI-like motif in Hip1R implicated in CLCa binding in vitro.

The manuscript characterises the Myosin-VI/Cathrin interaction, providing molecular information on the binding interface and on the functional implications for clathrin activities in polarised epithelial cells. The findings are novel and well presented, and uncover an interesting isoform-specific function of Myosin-VI that will be interesting for scientists in the field. However, a few issues remain open on the biological relevance of the interaction. I am in favour of publication in Nature Communications provided that these issues are addressed, as outlined below.

R. We thank the Reviewer for noting the novelty and importance of our work.

1. To evaluate the relevance of the Myosin-VI-long/CLCa interaction the authors generate Caco-2 cell lines stably expressing CLCa-WT and the I54D mutant. Analysing three-dimensional cysts grown with these cell lines ablated of endogenous CLCa, they observe morphologically defective clathrin coated pits but no polarity defects. Here some controls are needed.

R. We thank the Reviewer for having pointed out the need for additional controls. We have now better characterized the polarity of the different cysts as discussed in detail below.

- In Fig 5c the authors show an immuno-blot of RFP-CLCa in the cell lines used for the rescue experiments. It would be informative to show the relative levels of endogenous CLCa and ectopically expressed rescue constructs in polarised Caco-2 cells in an IB with the same exposure. If all exposures are already equal, the authors should comment on the fact that the rescue constructs are expressed at lower levels than the endogenous protein (and possibly repeat experiments with higher rescue levels), and that CHC levels seems reduced upon CLC depletion...

R. We thank the Reviewer for drawing our attention to this ambiguity. The blot presented in (former) Fig. 5c (now Fig. 5a) is actually taken from the same filter exposure and indeed the rescue constructs are expressed at a much lower level. Honestly, we do not know the reason of the low expression of the RFP-CLCa constructs. We sorted the infected cells and selected cells for the highest expression. The same constructs in MEF cells expressed much better and replaced the endogenous protein even in the absence of siRNA KD (as visible below). Thus, this reduced expression seems to be strictly Caco-2 dependent. In any case the expression of the WT and I54D mutant (both in IB and IF) are very similar allowing us to make conclusions.

..and that CHC levels seems reduced upon CLC depletion.

R. This is an interesting observation. We analyzed recent literature to find that this effect is visible with siRNA oligos different from our and in various cell lines (Majeed et al Nat Comm. 2014, Fig. 1d; Kim et al PLoS One 2011, Fig. 4J). In addition, we repeated the siRNA experiment with two oligos in HeLa cells and we

measured CHC expression by IB. Also in these cases, we found a reduction in CHC expression upon CLCa but not CLCb depletion (new Supplementary Fig. 11b-c). We added a comment in the legend as this phenomenon, to our knowledge, has not been reported previously. What is relevant for our conclusions is that partial CHC reduction is apparently not responsible for the phenotype as the effect can be rescued by WT CLCa but not mutant CLCa.

- in Fig 5d, representative images of cysts grown with ctrl and Caco-2 rescue cell lines are shown. The authors conclude that no polarity nor morphogenetic defects are visible. However *E-cadh* seems more apical in the rescue lines, both the one expressing CLCa-WT and the one expressing the I54D mutant. In addition, cysts with these cell lines show multilayering and cells inside the lumen. The authors should comment on this evidence, and stain the cysts with additional polarity markers (for instance *aPKC*, *actin*, *DLG*...) to ensure that apico-basal polarity is properly rescued and preserved in the I54D mutant cells.

R. We thank the reviewer for bringing this to our attention. We apologize for the poor choice of images. We have now performed additional IF and EM experiments that confirmed that apico-basal polarity is properly rescued and preserved in the I54D mutant cysts. No difference between wt and mutant cysts are visible in terms of multilayering or cells inside the lumen. We changed the panel (new Fig. 5b) and added few panels to the supplement (new Supplementary Figure 10) to better describe this point. Additional panels are here reported for the Reviewer's critique.

It would also be interesting to monitor the localisation of Myosin-VI in the cysts with CLCa rescue constructs.

R. Agree, unfortunately the localization of myosin VI in the cysts is not homogeneous and this depends mainly on the variability of expression of the CLCa rescue constructs. We saw a tendency for myosin VI to be more diffused in CLCa mutant cysts, but since this will be difficult to quantify we prefer not to risk overinterpreting our data.

- In Fig 5e, EM analysis of Caco-2 cysts show that the morphology of CCPs is impaired in the CLCa-I54D mutant. In these experiments the lumen and the apical side of Caco-2 cells is visualised in cysts treated with cholera toxin, which induces lumen expansion and might affect membrane tension. It would be useful to repeat the experiment in cysts without CTX treatment.

R. We are not totally sure to have understood this point. In any case we tried the experiment suggested. Unfortunately, Caco-2 cysts prepared without CTX treatment did not give rise to fully mature cysts, as previously reported (Jaffe et al, JCB 2008). After 2 weeks in culture, most of the cysts are still without lumen. Moreover, qPCR analysis showed that expression of CLCs came back to the original state indicating that siRNA oligos are not effective any more. Data are reported below.

- Also, if possible, it would be important to know if the defective morphology observed in the CCPs of cysts with CLCa-I54D results in defective endocytosis, or can be tolerated / bypassed.

R. This is an interesting point that is difficult to establish though, as it requires dynamic measurement of internalization in 3D.

In vivo endocytosis assays of specific cargos in 3D cysts has never been established. Ideally, we would need a KI cell line in which the CLCa mutation is included in the alleles of the gene. When we started the rescue experiments, we thought to use a CRISPR/CAS9 approach and we made use of two constructs to edit the CLCs genes generously provided by our colleague Sandy Schmid (Chen PH, Dev Cell 2017). Unfortunately, we failed to obtain edited cells even evaluating hundreds of clones FACS-sorted. This may also depend on the fact that Caco-2

suffer if cultured as single cells – a procedure needed for the CRISPR approach. The failure of this approach has greatly reduced the number of feasible experiments as we are stuck with transient depletion.

2. In Fig 6, the authors identify an amino acidic stretch (IENDEA) in Hip1R partly overlapping with the CLCa-binding region of Myosin-VI. This stretch binds to CLCa *in vitro* and is out-competed by Myosin-VI-1050-1131. Notably, the IENDE motif is also present in CLCb and mediate binding of Hip1R to CLCb, possibly with higher affinity based on Supplementary Fig. 10h. The relevance of the competitive interaction between Hip1R and Myosin-VI in binding to CLCa for clathrin-mediated endocytosis is not worked out further in cells. Thus, it remains speculative. The authors should analyse the interplay between Myosin-VI and Hip1R in CLC-a/b binding in cells, or move this information to the supplementary material.

R. We performed the suggested experiment using highly polarized Caco-2 cells in 2D culture as we could not use 3D cysts for biochemical assays.

Co-immunoprecipitation with RFP-CLCa confirmed once again that I54D is impaired in binding with myosin VI, while showing no difference between CLCa WT and mutant for Hip1R whose binding was minimal (shown below). This is not surprising considering the low affinity interaction that we have now measured by ITC (new Fig. 6d). Thus, the relevance of the competitive interaction *in vivo* remains to be established and possibly requires high resolution imaging with the complexities we described above in the response to Reviewer 2.

We are willing to move the data and the model to supplementary material if required, although we propose to leave it in the main figures as they suggest an interesting and novel scenario in terms of regulation.

Minor points

- Experimental errors bars should be added to FP graph shown in Fig. 2c-2d and Supplementary Fig 9b.

R. We apologize for this misunderstanding. The FP graphs show a single representative experiment. In a single experiment, duplicates have the same starting concentration of myosin VI that is accurately measured at the end of each experiment. Between different experiments (>3/type), the concentration of myosin VI varies depending on the technical error of pipetting and the status of

the purified proteins, making it impossible to calculate the SD for single points. Thus, based on the raw data, we obtained a Kd value for each experiment and we calculated SD and SE for the average of Kds that represent the experimental variability. We added the raw data of the measurements in the source data file for further inspection.

- To make the structure of the Myosin-VI/CLCa-46-61 easier to understand, the authors should consider illustrating the structure with same colors of the cartoons in Fig 1a-2a, and with orientations related to one another, including the enlarged views. Also, it might be beneficial to organise the panels in order to have the validation of the interface shown in Fig 5a-5b included in Fig. 4 together with the close-up views.

R. We are very grateful for bringing this to our attention. We have now changed the colors of the cartoons to maintain the color-code of the structures. We specified the orientation of the structures relative to each other and wherever possible displayed structures in the same orientation for clarity. We also re-ordered the panels such that the validation of the interface appears together with structural details.

- In supplementary Fig 1a no asterisk is visible. The figure or the caption should be modified to account for this discrepancy.

R. Agree. We have revised the Figure accordingly.

- Molecular Weight Marker should be added in the blot/SDS-PAGES presented in Supplementary Fig 2d.

R. Agree. We have revised the Figure accordingly.

- In Fig 1 legend and supplementary Fig 9a, Myosin-VI-short seems to have the same numbering of Myosin-VI-long (ending at residue 1131). This is somehow confusing, it would be better to modified it.

R. We apologize for the confusion. This way to describe the constructs was suggested to us by a Reviewer during the revision of our previous paper (Wollscheid et al. 2016) and we would prefer to stick with the same modality. Nonetheless, we have now changed Fig. 1a and Suppl. Fig. 9a to better described the various constructs used.

- Supplementary Fig 10f shows the KO efficiency of siRNA targeting CLCa/b. Based on the plots, in cysts the KO-efficiency after 7 days is about 60% compared to CTRL cysts, which is not very high. The authors should comment on these data, and on the choice of using transient silencing for the cysts experiments.

R. As mentioned before, we tried with no success to use a KO approach. We performed a third experiment that confirmed previous data and showed a better KD efficiency (CLCa: 79% for KD, 73% for WT and 80% for I54D), now shown in the new Supplementary Fig. 11a.

Reviewers' Comments:

Reviewer #2:

Remarks to the Author:

The revised manuscript has addressed my comments from the first review. However, there are still some issues:

In the new diagram of myosin VI in fig. 1A: The label 'motor head' is a misnomer. The 'head' is formed by the motor domain and the neck (or lever), and thus the label 'motor head' does not make sense. This needs to be changed to motor. Moreover, the schematic is missing both the 3-Helix bundle, and the single alpha helical (SAH) domain (incidentally also nicely and recently studied using NMR – see JACS, 2019, Barnes et al., and see references therein) – residues 918 – 985 of Pig myosin 6, which is also expected to form part of the lever, along with the IQ-calmodulin binding region.

Fig 5C is missing a scale bar for the low power images at the top. I also realise that I assumed that this was a low power EM image? But the legend is unclear. Is this just a brightfield image from a light microscope? Please describe in the legend.

In Fig. 6e, what are the different domains of myosin VI as shown in the diagram – I assume the 'boxing glove' like structure is the motor domain, the white region the single IQ motif? What then follows? How is the green circle at the C-terminus related back to the diagram in Figure 1A. This 'domain' should also be indicated in Figure 1A, for clarity.

It's good that the authors have added to the clathrin pit data (Fig 5, Supplemental Fig. 11)

However, there is a misspelling of length on the axis of Fig. 11d (which is number of pits per length of apical PM in microns. However, I'm confused by the legend which says 'per 100 microns'. As the numbers are very low - 0.03 is the maximum, this would mean that pits are very sparse! Less than 1 per mm!. The data in supplemental fig 11 would also be better represented by box and whisker or violin plots – demonstrating the range of the data more clearly. Moreover, it still does not say how many cells were analysed. It does say that there were three separate experiments only. Is the mean +/- SEM a mean of 3 experiments? or this is a mean of all the measurements? This is important for the statistical analysis. I realise that these are probably non-trivial measurements to make, but they should really be given as numbers per cell, and then an average number for however many cells were counted, which is then used in the statistical analysis.

Finally, I checked the de Jonge JJ review in Febs Lett, re the dimerization of myosin VI. The way in which myosin VI is shown to be 'dimerised' by DAB2 in that paper (Fig. 1), is shown differently to the model in this paper in Fig 6E. In the de Jonge paper, Dab2 is shown as a dimer, which then binds two molecules of myosin VI and the legend to this figure states 'two Dab2 monomers may dimerize MYO6'. I think one has to be careful what one means by dimerization here – because what is really happening is 'molecular crowding' as the monomers bind to their binding partners, the myosin VI itself is not a dimer. Thus, the model in Fig. 6E is misleading, as there is only one molecule of Dab2 shown per myosin 'dimer', and the myosin VI appears to be dimerising through its tail domain and part of the upstream sequence, which is not the case. Thus the model shown in Fig 6E needs revising.

Reviewer #3:

Remarks to the Author:

The authors have addressed all my concerns, discussing the critical points and adding new experiments. Therefore I fully support the publication of the manuscript in Nature Communications.

I leave to the Editors the decision whether to include the data describing the competitive interaction between Hip1R and Myosin-VI in the main figures as putative scenario, or in the Supplementary Material.

I add a note regarding the analyses of CCPs at the apical site in Caco-2 cysts. What I meant in my previous comment is that CTX treatment enlarges the lumen of the cysts artificially, resulting in an

unphysiological stretching of the apical membrane that might impact on membrane trafficking and endocytosis. Based on figure 3 of the Jaffe paper cited by the authors (Jaffe, JCB 2008), it should be possible to visualise the apical membrane in Caco-2 cysts without CTX treatment, though it will be indeed more difficult.

Reviewer #2:

The revised manuscript has addressed my comments from the first review. However, there are still some issues:

In the new diagram of myosin VI in fig. 1A: The label 'motor head' is a misnomer. The 'head' is formed by the motor domain and the neck (or lever), and thus the label 'motor head' does not make sense. This needs to be changed to motor. Moreover, the schematic is missing both the 3-Helix bundle, and the single alpha helical (SAH) domain (incidentally also nicely and recently studied using NMR – see JACS, 2019, Barnes et al., and see references therein) – residues 918 – 985 of Pig myosin 6, which is also expected to form part of the lever, along with the IQ-calmodulin binding region.

R. We apologize for the misnomer. We modified the figure adding all the helical domains cited together with the related papers in the figure legend.

Fig 5C is missing a scale bar for the low power images at the top. I also realise that I assumed that this was a low power EM image? But the legend is unclear. Is this just a brightfield image from a light microscope? Please describe in the legend.

R. Yes. They are indeed brightfield images taken on an Evos microscope. This information is now reported in the figure legend and the missing scale bar is now included in the figure.

In Fig. 6e, what are the different domains of myosin VI as shown in the diagram – I assume the 'boxing glove' like structure is the motor domain, the white region the single IQ motif? What then follows? How is the green circle at the C-terminus related back to the diagram in Figure 1A. This 'domain' should also be indicated in Figure 1A, for clarity.

R. We thank the reviewer for drawing our attention to this lack of clarity. We found it quite difficult to depict all of the structural details of myosin VI in the cartoon of Fig. 6e. As mentioned before, we have now modified also Fig. 1a in order to maintain the same color code. The 'boxing glove' like structure is the motor domain, the white region is the single IQ motif, the region that follows is the remaining helical tail which includes 3HB and SAH while the green is the globular tail. This information is now included in the figure legend.

It's good that the authors have added to the clathrin pit data (Fig 5, Supplemental Fig. 11)

However, there is a misspelling of length on the axis of Fig. 11d (which is number of pits per length of apical PM in microns. However, I'm confused by the legend which says 'per 100 microns'. As the numbers are very low - 0.03 is the maximum, this would mean that pits are very sparse! Less than 1 per mm!. The data in supplemental fig 11 would also be better represented by box and whisker or violin plots – demonstrating the range of the data more clearly. Moreover, it still does not say how many cells were analysed. It does say that there were three separate experiments only. Is the mean +/- SEM a mean of 3 experiments? or this is a mean of all the measurements? This is important for the statistical analysis. I realise that

these are probably non-trivial measurements to make, but they should really be given as numbers per cell, and then an average number for however many cells were counted, which is then used in the statistical analysis.

R. We thank the reviewer for making us aware of the discrepancy in the figure legend (which was wrong) of Supplemental Fig. 11d. The number of pits are normalized to the apical PM length and expressed as number/microns. The number of cellular profiles (previously reported as N in the figure) is now also indicated in the legend. As pointed out by the reviewer, this analysis is non-trivial as clathrin-coated pits at the apical surface are scarce. Thus, data obtained from the 3 independent experiments were pooled together for the statistical analysis. We also added to the figure the average number of coated pits per cell, as requested. We hope that these modifications will better clarify our morphometrical analysis.

Finally, I checked the de Jonge JJ review in Febs Lett, re the dimerization of myosin VI. The way in which myosin VI is shown to be 'dimerised' by DAB2 in that paper (Fig. 1), is shown differently to the model in this paper in Fig 6E. In the de Jonge paper, Dab2 is shown as a dimer, which then binds two molecules of myosin VI and the legend to this figure states 'two Dab2 monomers may dimerize MYO6'. I think one has to be careful what one means by dimerization here – because what is really happening is 'molecular crowding' as the monomers bind to their binding partners, the myosin VI itself is not a dimer. Thus, the model in Fig. 6E is misleading, as there is only one molecule of Dab2 shown per myosin 'dimer', and the myosin VI appears to be dimerising through its tail domain and part of the upstream sequence, which is not the case. Thus the model shown in Fig 6E needs revising.

R. We thank the Reviewer for pointing out this subtle but important point. We revised the figure accordingly.

Reviewer #3:

I add a note regarding the analyses of CCPs at the apical site in Caco-2 cysts. What I meant in my previous comment is that CTX treatment enlarges the lumen of the cysts artificially, resulting in an unphysiological stretching of the apical membrane that might impact on membrane trafficking and endocytosis. Based on figure 3 of the Jaffe paper cited by the authors (Jaffe, JCB 2008), it should be possible to visualise the apical membrane in Caco-2 cysts without CTX treatment, though it will be indeed more difficult.

R. Fig. 3 of Jaffe et al was meant to show that the apical surface is established at the beginning of the process by taking advantage of aPKC as an IF marker. Thus, it is true that it might be possible to visualize the apical membrane without CTX treatment by IF. What is extremely difficult (if not impossible) is to perform a statistical analysis of CCPs in this condition.

We looked by EM at the cysts prepared without CTX and we could confirm that the vast majority of them are without the lumen (below to 10%, in good accordance with Jaffe et al.). Moreover, the qPCR analysis (presented in the

previous reply) showed that expression of CLCs came back to the original state indicating that siRNA oligos are not effective any more after 12 days.

Thus, to analyze the behavior of the few CCPs at the apical surface we should first identify by EM CLC/CLCb negative cysts (if any) among the 10% empty fully mature cysts (by immunostaining). Considering the scarce number of pits present at the apical surface we regret to say that this kind of analysis is impossible to pursue in a reasonable time. We further note that even if CTX causes unnatural stretching beyond normal luminal stretching, we are comparing CCPs with mutant and wild-type CLCs under the same conditions and find that the mutation shows a phenotype that can be rescued by the WT construct, so the conclusions attributing a role for myosin-CLC interaction in CCP formation at the lumen are valid.